# TMEM33 regulates intracellular calcium homeostasis in renal tubular epithelial cells

Malika Arhatte[1,13], Gihan S. Gunaratne[2,13], Charbel El Boustany[1], Ivana Y. Kuo [3], Céline Moro[1], Fabrice Duprat [1], Magali Plaisant[1], Hélène Duval[1], Dahui Li[4], Nicolas Picard [5], Anais Couvreux[1], Christophe Duranton[6], Isabelle Rubera[6], Sophie Pagnotta[7], Sandra Lacas-Gervais[7], Barbara E. Ehrlich [8], Jonathan S. Marchant [9], Aaron M. Savage[10,11], Fredericus J.M. van Eeden[11,12], Robert N. Wilkinson [10,11], Sophie Demolombe[1], Eric Honoré[1,14] & Amanda Patel [1,14]

Mutations in the polycystins cause autosomal dominant polycystic kidney disease (ADPKD). Here we show that transmembrane protein 33 (TMEM33) interacts with the ion channel polycystin-2 (PC2) at the endoplasmic reticulum (ER) membrane, enhancing its opening over the whole physiological calcium range in ER liposomes fused to planar bilayers. Consequently, TMEM33 reduces intracellular calcium content in a PC2-dependent manner, impairs lysosomal calcium refilling, causes cathepsins translocation, inhibition of autophagic flux upon ER stress, as well as sensitization to apoptosis. Invalidation of TMEM33 in the mouse exerts a potent protection against renal ER stress. By contrast, TMEM33 does not influence *pkd2*-dependent renal cystogenesis in the zebrafish. Together, our results identify a key role for TMEM33 in the regulation of intracellular calcium homeostasis of renal proximal convoluted tubule cells and establish a causal link between TMEM33 and acute kidney injury.

[1] Université Côte d'Azur, Centre National de la Recherche Scientifique, Institut national de la santé et de la recherche médicale, Institut de Pharmacologie Moléculaire et Cellulaire, Labex ICST, Valbonne 06560, France. [2] Department of Pharmacology, University of Minnesota, Minneapolis, MN 55455, USA. [3] Cell and Molecular Physiology Stritch School of Medicine Loyola University Chicago, Maywood, IL 60153, USA. [4] The State Key Laboratory of Pharmaceutical Biotechnology, Department of Pharmacology and Pharmacy, The University of Hong Kong, Hong Kong SAR, China. [5] Laboratory of Tissue Biology and Therapeutic Engineering, UMR 5305 CNRS, University Lyon 1, Lyon 69367, France. [6] Université Côte d'Azur, LP2M CNRS-UMR7370, Labex ICST, Medical Faculty, Nice 06108, France. [7] CCMA UFR Sciences, Université Côte d'Azur, Nice 06108, France. [8] Department of Pharmacology, School of Medicine, Yale University, New Haven, CT 06520-8066, USA. [9] Department of Cell Biology, Neurobiology and Anatomy, Medical College of Wisconsin, Milwaukee, WI 53226, USA. [10] Department of Infection, Immunity and Cardiovascular Disease, Medical School, University of Sheffield, Sheffield S102TN, UK. [11] The Bateson Centre, University of Sheffield, Sheffield S102TN, UK. [12] Department of Biomedical Science, University of Sheffield, Sheffield S102TN, UK. [13] These authors contributed equally: Malika Arhatte, Gihan S. Gunaratne. [14] These authors jointly supervised this work: Eric Honoré, Amanda Patel. Correspondence and requests for materials should be addressed to E.Hé. (email: honore@ipmc.cnrs.fr)

Mutations in the polycystin genes *Pkd1* (encoding polycystin-1; PC1) and *Pkd2* (encoding PC2) cause autosomal dominant polycystic kidney disease (ADPKD), the most common monogenic disease[1]. This is a multisystemic disease associated with the development of focal cysts in the kidney, liver and pancreas, as well as arterial structural anomalies and hypertension. A two hit mechanism was proposed including one inactivating germinal mutation and an additional event affecting the level of expression of the second allele (somatic inactivating mutation or a hypomorphic dosage effect)[1]. PC2 is a member of the Transient Receptor Potential (TRP) ion channel family (also called TRPP2) made of six transmembrane segments with a pore (P) domain located between S5 and S6[2,3].

PC2 is targeted to the primary cilium and its ion channel function within this tiny organelle protruding at the apical side of tubular epithelial cells was recently demonstrated using patch clamp recordings[4,5]. Ciliary PC2 of mouse inner medullary collecting duct cells mainly conducts monovalent cations, as well as calcium, is inhibited at negative potentials by high external calcium concentration (IC$_{50}$: 17 mM), but stimulated by a rise in intracellular calcium (EC$_{50}$: 1.3 μM)[4,5].

PC2 is also retained in the endoplasmic reticulum (ER) through a retention signal in its carboxy terminal domain[6,7]. PC2 was shown to act as a calcium releasing channel activated by cytosolic calcium (calcium-activated calcium release) at the ER membrane[7]. An EF-hand domain in the cytoplasmic C terminus is proposed to underlie activation of PC2 by cytosolic calcium[7–11]. Single channel recordings of microsomes enriched ER PC2 fused in planar lipid bilayers show a bell-shaped dependence on cytoplasmic calcium, with a maximum opening at 0.3 μM Ca$^{2+}$[7,10]. Additional findings indicate that PC2 interacts with the type I IP$_3$R to modulate intracellular calcium signaling[12,13]. Calcium flowing through the IP$_3$R is thought to locally activate PC2, thus amplifying calcium release from the ER[12,13]. Accordingly, calcium transients elicited by vasopressin in LLC-PK1 cells were greatly enhanced and prolonged when PC2 was overexpressed[7,10]. Conversely, PC2 was also shown to lower ER calcium concentration resulting in decreased IP$_3$-dependent responses[14]. ER-resident PC2 counteracts the activity of the calcium ATPase by increasing passive calcium leak[14]. Accordingly, knock down of PC2 in renal epithelial cells increases ER calcium content[14]. However, a role for PC2 in ER calcium leak remains controversial[12]. Thus, depending on the gating mode (calcium-gated or leak) PC2 differentially influences IP$_3$-dependent responses[7,14]. What regulates PC2 gating at the ER is currently unknown.

In the present report, we demonstrate in renal proximal convoluted tubule (PCT) cells, that the ER conserved transmembrane protein TMEM33 interacts with PC2, enhancing its channel activity over the whole physiological cytosolic calcium range in ER liposomes fused to planar bilayers. Finally, we establish a functional link between TMEM33 and acute kidney injury (AKI), while *Pkd2*-dependent cystogenesis is independent of TMEM33.

## Results

**TMEM33 is in a complex with PC2 at the ER of PCT cells**. Our previous proteomic findings indicated a possible biochemical interaction between PC2 and TMEM33[15]. TMEM33 was co-purified with PC2 (73 peptides identified versus 0 peptide in control mock conditions), together with IP$_3$R3 and IP$_3$R1[15]. When the C-terminal domain of PC2 was truncated (PC2–742X and PC2–690X), the number of identified TMEM33 peptides dropped (7 and 6 peptides, respectively). Co-immunoprecipitation experiments performed in PCT cells confirmed that TMEM33 is found in a complex together with PC2 (Fig. 1a, b, Supplementary

Fig. 1a). PC2 was specifically immunoprecipitated with both TMEM33-HA and HA-TMEM33 in stably complemented TMEM33$^{−/−}$ cell lines conditionally expressing TMEM33 at levels (TMEM33-HA/TOP1 and HA-TMEM33/TOP1: 0.55 and 0.15, respectively when induced with DOX), comparable to its native physiological expression in WT PCT cells (TMEM33/TOP1: 0.2) (Supplementary Fig. 1a). Importantly, immunoprecipitation of TMEM33-HA or HA-TMEM33 with native PC2 was specifically observed with an anti-HA antibody, and not with control rat IgG (Supplementary Fig. 1a). Again partial truncation of the PC2 C-terminal domain lowered the interaction with TMEM33 (Fig. 1a, b). Next, we used the LexA/B42 based Grow'n'Glow yeast-2 hybrid system to further confirm the interaction (direct or indirect) between TMEM33 and PC2 (Fig. 1c). Interaction (i.e., yeast growth) was observed between the C terminus of TMEM33 and the N terminus of PC2, as well as between the N terminus of TMEM33 and the C terminus of PC2 (Fig. 1c). Of note, TMEM33 N and C termini (as well as the yeast homologues) are predicted to be both facing the cytosolic side[16–18], although an alternative topology was also proposed[19]. Thus, TMEM33 establishes a complex molecular interaction with PC2 in renal tubular epithelial cells. In PCT cells, we observed an obvious co-localization of TMEM33-GFP with mCherry-PC2 at the ER membrane, using ER tracker as a specific marker for the ER (Fig. 1d). HA-TMEM33 visualized in the complemented TMEM33$^{−/−}$ PCT cell line was also co-localized with native PC2 at the ER (visualized by calnexin staining; Supplementary Fig. 5a). Importantly, co-localization was absent at the primary cilium (Fig. 1e). PC2 (red fluorescence, alexa 594) is present at the primary cilium (deep red fluorescence, alexa 647), while in the same cells TMEM33 (EGFP) is absent (although it is visible at the ER; Fig. 1d, e, Supplementary Fig. 5a).

TMEM33 mRNA, as detected by qPCR, is broadly expressed in mouse tissues (as previously reported for PC2[20]), including the kidney (Supplementary Fig.1b). Abundance of TMEM33 in renal tubule cells was confirmed using a TMEM33 LacZ reporter mouse line (Supplementary Fig. 1c).

Thus, TMEM33 is in a molecular complex with the ion channel PC2 at the ER membrane, but is absent from the primary cilium. In this report, we focused on the physiopathological function of TMEM33 in the kidney. In the next section, we investigated whether TMEM33 influences the regulation of intracellular calcium homeostasis through its interaction with PC2.

**TMEM33 diminishes IP$_3$-dependent calcium transients**. We investigated the effect of TMEM33 knock-down on the release of intracellular calcium in PCT cells. IP$_3$-dependent ER calcium release induced by purinergic stimulation in the absence of extracellular calcium was significantly enhanced when cells were transfected with two different validated siRNAs directed against TMEM33 (Fig. 2a, Supplementary Fig. 2a, Supplementary Fig. 2f, g, Supplementary Fig. 3c, d). TMEM33 or PC2 knock-down or knockout experiments were validated both at the mRNA (qPCR experiments) and protein levels (Western blots)(Supplementary Fig. 1d and 1g, Supplementary Fig. 2f-j). Of note, knock-down of TMEM33 did not affect PC2 protein expression, and vice-versa (Supplementary Fig. 2f-j). Conversely, TMEM33 overexpression had the opposite effect (Fig. 2b). Accordingly, the capacitative calcium entry induced by re-addition of extracellular calcium during ATP stimulation was enhanced when TMEM33 was knocked down, but lowered when TMEM33 was overexpressed (Fig. 2a, b). Regulation of intracellular calcium homeostasis was further studied in a conditionally complemented TMEM33$^{−/−}$ PCT cell line (Fig. 2c, d). In this cellular model, we used a Tet-on system allowing an inducible expression of TMEM33 (or TMEM33-HA or HA-TMEM33 tagged) along with a Cherry

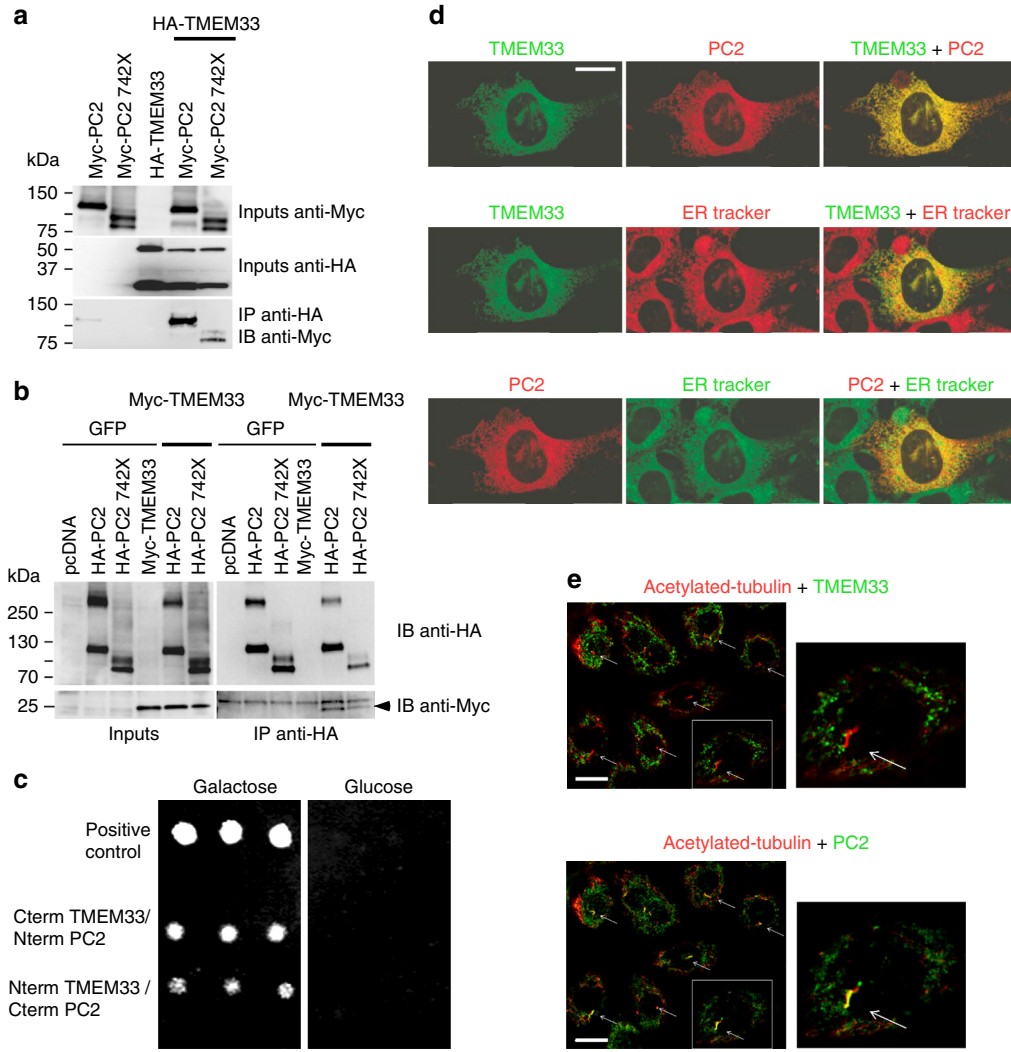

**Fig. 1** TMEM33 interacts with PC2 at the ER membrane of PCT cells. **a** Co-immunoprecipitation of TMEM33 with PC2 in PCT cells when TMEM33 is pulled down. A pathologic PC2 deletion mutant PC2–742X was also tested. **b** Same, but when PC2 is pulled down. Arrowhead indicates TMEM33 and the upper band is non-specific. **c** Yeast two hybrid experiments indicating an interaction (direct or indirect) between TMEM33 and PC2. We used a yeast strain with chromosomal LEU2 reporter gene harboring LexA operator binding sites. As a second reporter we used GFP under the control of the LexA operator (URA). The bait was either the COOH or NH2 terminus of TMEM33 expressed in frame with LexA (HIS), and the prey was either the COOH or NH2 terminus of PC2 expressed in frame with B42. As a further selection the prey product is only expressed in the presence of galactose. For a positive interaction, the yeast need to grow on medium -HIS-TRP-URA-LEU + Galactose and light up positive for GFP. As a positive control we used LexA-p53 with B42-LTA. As negative controls, we either expressed the LexA-CtermTMEM33 with the empty prey vector or B42-CtermPC2 with the empty bait vector. No growth was observed with any combination of either prey alone or bait alone. **d** Top: Co-localization of TMEM33-GFP (green) and mCherry-PC2 (red) in PCT cells; Middle: co-localization of TMEM33-GFP (green) with ER tracker as an ER marker (red); Bottom: co-localization of mCherry-PC2 (red) with ER tracker (green). The merged image (yellow shows co-localization) is shown at the right. False colors were used to consistently illustrate co-localizations in yellow (ER tracker blue). The scale bar corresponds to 20 μm. **e** TMEM33 is absent at the primary cilium, unlike PC2. Acetylated-tubulin in red labels the primary cilia and TMEM33 (top panels) or PC2 (bottom panels) are labeled in green (false colors). Co-localization of PC2 and acetylated tubulin is seen in yellow (merged image; bottom panels). The right panels are magnification of the boxed area shown in the left panels. Arrows indicate primary cilia. The scale bar corresponds to 15 μm. Source data are provided as a Source Data file

fluorescent reporter in the presence of doxycycline (DOX)(Supplementary Fig. 2e). Control cells were the parental $TMEM33^{-/-}$ PCT cell expressing the repressor together with a CD8 construct retained in the ER (CD8ER)[21] and treated with DOX, similarly to the cell line expressing TMEM33 (Fig. 2d). We isolated a clone with an inducible TMEM33 expression (i.e. in the presence of DOX) comparable to the level found in the WT PCT cells (Fig. 2c). Again, when TMEM33 expression was enhanced by DOX addition, the peak ATP response elicited in the absence of extracellular calcium was blunted, as well as the associated capacitative calcium entry (Fig. 2c). Importantly, these effects

were absent in the control CD8ER complemented $TMEM33^{-/-}$ PCT cell line (Fig. 2d). Strikingly, in PCT $Pkd2^{-/-}$ cells TMEM33 knock-down or overexpression failed to affect the ATP response (Fig. 2e, f, Supplementary Fig. 1d, Supplementary Fig. 2b). These findings indicate that TMEM33 impacts the regulation of intracellular calcium homeostasis through PC2.

The SERCA inhibitor thapsigargin does not allow the discrimination of selective changes in ER calcium content or number/activity of ER leak calcium channels since both parameters are linked[14]. However, the calcium ionophore ionomycin in the absence of extracellular calcium allows the

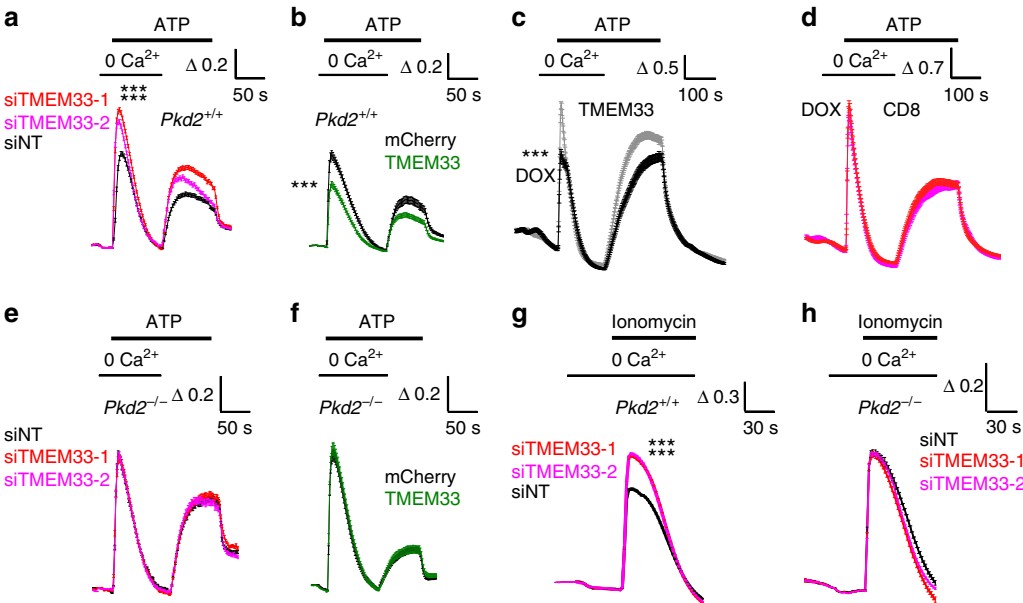

**Fig. 2** TMEM33 regulates intracellular calcium homeostasis. **a** Increases in cytosolic calcium concentration are expressed as ratios of 340:380 nm fluorescence signals ($\Delta R/R_O$). $\Delta R$ is the fluorescence ratio (340 nm/380 nm) measured at a given time divided by the initial ratio at time 0 ($R_O$). Transfection of PCT cells with two siRNAs directed against TMEM33 increases ATP calcium transients recorded in the absence of extracellular calcium, as compared to the control non-targeting siRNA condition (siNT, $n = 684$, black symbols; siTMEM33-1, $n = 589$, red symbols; siTMEM33-2, $n = 649$, magenta symbols). Re-addition of extracellular calcium during the ATP stimulation elicits a capacitive calcium entry. **b** Same in PCT cells transiently overexpressing TMEM33 or not (Cherry, $n = 244$, black symbols; TMEM33, $n = 254$, green symbols). **c** Same in a conditional PCT cell line expressing TMEM33 when induced with DOX (black trace; $n = 321$) or not (gray trace; $n = 464$). **d** Same in a conditional PCT line expressing CD8ER in the presence of DOX (magenta trace; $n = 181$) or not (red trace; $n = 266$). **e** Same in $Pkd2^{-/-}$ PCT cells transfected or not with siRNAs against TMEM33 (siNT, $n = 764$; siTMEM33-1, $n = 1025$; siTMEM33-2, $n = 856$). **f** Same in $Pkd2^{-/-}$ PCT cells overexpressing or not TMEM33 (Cherry, $n = 144$; TMEM33, $n = 141$). **g** Ionomycin calcium transients in PCT cells ($Pkd2^{+/+}$) bathed in the absence of extracellular calcium and transfected or not with siRNAs against TMEM33 (siNT, $n = 1648$; siTMEM33-1, $n = 11615$; siTMEM33-2, $n = 1587$). **h** Same in $Pkd2^{-/-}$ PCT cells (siNT, $n = 627$; siTMEM33-1, $n = 769$; siTMEM33-2, $n = 659$). Values are means ± SEM. One star indicates $p < 0.05$, two stars $p < 0.01$ and three stars $p < 0.001$, with a Student's $t$ test used to evaluate statistical significance. Source data are provided as a Source Data file

precise measurement of stored intracellular calcium content[14]. Notably, TMEM33 knock-down significantly increased the release of calcium from intracellular stores induced by ionomycin in a PC2-dependent manner (Fig. 2g, h). A similar finding was obtained with the conditional TMEM33 cell line, although not in the parental CD8 expressing TMEM33$^{-/-}$ cell line (Supplementary Fig. 3a, b). Thus, our findings indicate that TMEM33 controls intracellular calcium homeostasis through PC2. Next, we investigated whether TMEM33 affects the gating of ER PC2.

**TMEM33 stimulates PC2 calcium-dependent activity**. PC2 is strongly upregulated in both acute and chronic kidney diseases[22–24]. In light of these findings, we investigated the effect of PC2 overexpression in PCT cells. When PC2 was transiently overexpressed, an increase in basal cytosolic calcium was consistently observed (Fig. 3a). Moreover, PC2 overexpression mildly reduced the peak ATP response (Fig. 3b). Notably, in this condition when TMEM33 was knocked down, the basal level of cytosolic calcium became normalized (Fig. 3a). Again, in PCT cells overexpressing PC2, an increase in the ATP response was detected upon TMEM33 knock-down (Fig. 3b). Similar findings were obtained with a PCT cell line stably overexpressing PC2 (Supplementary Fig. 3c, d). These results are consistent with a stimulation of PC2 opening by TMEM33. To directly test this idea, we fused ER enriched microsomes of PCT cells overexpressing PC2 in planar lipid bilayers and electrophysiologically recorded channel activity[7]. Ba$^{2+}$ was used as the charge carrier on the *trans* (luminal) side, as it does not affect channel activity

on either side of the channel and permeates PC2[7]. The open channel probability showed a bell-shaped dependence on cytoplasmic calcium with a maximum observed at 0.3 μM, as previously reported[10] (Fig. 3c, d). When TMEM33 was co-expressed with PC2, we observed a prominent increase in channel open probability across the entire activating calcium range, with no significant change in single channel current amplitude (Fig. 3c, d). The decrease in channel activity expected at the higher calcium range with PC2 alone was absent when TMEM33 was overexpressed (Fig. 3d). Thus, TMEM33 modulates the gating of PC2 by cytosolic calcium, removing channel inactivation at the higher cytosolic calcium concentrations.

IP$_3$R activation drives the calcium refilling of other cellular organelles, including lysosomes[25–27]. Next, we investigated whether TMEM33 might indirectly influence endolysosomal calcium content because of an altered IP$_3$ calcium signaling.

**TMEM33/PC2 impacts endolysosomal structure and function**. Both TMEM33-HA and HA-TMEM33, as well as native PC2 are found at the ER membrane (Fig. 1d, Supplementary Fig. 5a), but are also detected within lysosomes (Supplementary Fig. 5b, c). We found several lines of evidence that suggest a key role for the ER TMEM33/PC2 complex on endolysosomal structure and function. First, endosomes expressing Two-pore channel 1 (TPC1), as well as lysosomes expressing TPC2 were found in close proximity to the ER (Supplementary Fig. 4a, b). Second, we measured the amount of calcium stored in lysosomes using the lysosomotropic agent glycyl-L-phenylalanine 2-naphthylamide (GPN)

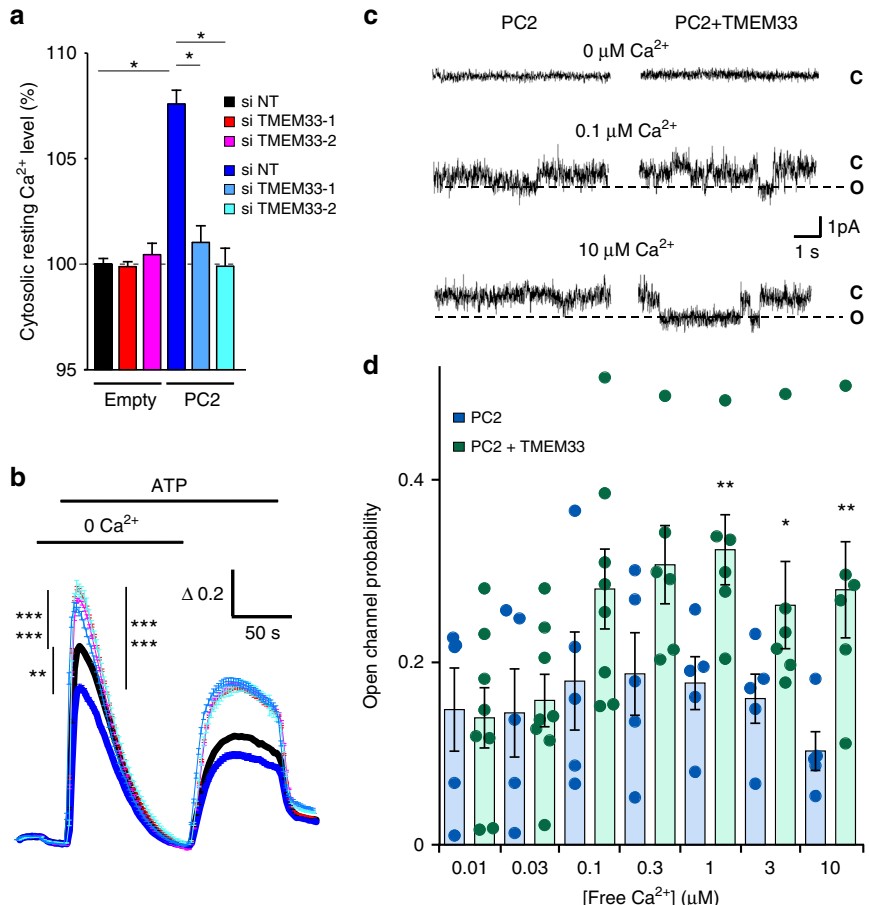

**Fig. 3** TMEM33 controls the calcium-dependent gating of PC2. **a** Increases in cytosolic calcium concentration are expressed as ratios of 340:380 nm fluorescence signals ($\Delta R/R_O$). Relative (to the siNT empty vector condition; black bar) basal cytosolic calcium levels in PCT cells transfected or not with siRNAs against TMEM33 (red and magenta bars). The effect of PC2 overexpression (as indicated at the bottom of the graph) was also investigated (dark blue bar, siNT and light blue bars: siTMEM33–1 and siTMEM33–2). Experimental points are available in the data source file. Numbers of cells analyzed are indicated in Fig. 2 and (**b**) legends. **b** ATP-induced calcium transients in the absence of extracellular calcium (siNT, $n = 2353$; siTMEM33–1, $n = 2437$; siTMEM33–2, $n = 2201$; siNT + PC2, $n = 262$; siTMEM33–1 + PC2, $n = 246$, siTMEM33 + PC2, $n = 322$). Same cells as in (**a**). **c** PC2 channel activity elicited with 0, 0.1 and 10 µM free calcium on the cis-side (cytosolic) in ER liposomes fused to planar bilayers (holding potential: 0 mV). As previously reported, the slope conductance of the channel was 85 pS when PC2 was activated by cytosolic calcium[7, 10]. C indicates the closed state and O the open state (dashed line). **d** Open channel probability of PC2 measured at increasing concentration of free cytosolic calcium (PC2: blue bars and PC2 + TMEM33: green bars). Values are means ± SEM overlaid with dot plots for (**d**). One star indicates $p < 0.05$, two stars $p < 0.01$ and three stars $p < 0.001$, with a Student's $t$ test for (**a**, **b**), as well as a Mann–Whitney test for (**d**) used to evaluate statistical significance. Source data are provided as a Source Data file

(Fig. 4a, b). Strikingly, TMEM33 significantly attenuated the amplitude of the GPN response in TMEM33 conditional PCT cells, unlike in parental $TMEM33^{-/-}$ PCT cells expressing CD8ER (Fig. 4a, b). Moreover, the effect of TMEM33 expression on lysosomal calcium load was suppressed when PC2 was knocked down (Fig. 4c, d). Third, TMEM33 expression produced an increase in the size of endolysosomes (Supplementary Fig. 4c–f). Fourth, TMEM33 conditional expression increased cytosolic, as well as extracellular cathepsins or N-acetyl-beta-D-glucosaminidase (NAG) (Supplementary Fig. 6). Cleavage of the fluorogenic cathepsin B/L substrate was detected when added to non-permeabilized cells, but not when added to cell-free conditioned media, suggesting that the extracellular activity of lysosomal cathepsins was due to membrane-associated proteases. Notably, the effect of TMEM33 on cathepsins or NAG translocation was markedly enhanced by tunicamycin (TM) treatment that causes ER stress (Supplementary Fig. 6). Remarkably, altered translocation of cathepsins or NAG upon TMEM33 expression was again blunted by knocking down PC2 (Fig. 4e-g; Supplementary Fig. 7). By contrast, overexpressing PC2 had the opposite effect (Fig. 4e-g; Supplementary Fig. 7).

Altogether these findings suggest that the decrease in IP3 signaling mediated by TMEM33 impacts lysosomal size and function in a PC2-dependent manner. Next, we investigated whether TMEM33/PC2 might also influence the lysosomal degradation pathway of autophagy.

**TMEM33 inhibits autophagic flux upon ER stress**. Recent studies indicate that PC2 stimulates autophagy in a variety of cell types, including renal epithelial cells, as well as cardiomyocytes, involving both the primary cilium and intracellular calcium stores[28–31]. LC3 is the most widely used autophagosome marker because the amount of LC3 II (conjugated to phosphatidyletha-nolamine) reflects the number of autophagosomes[32]. Degradation of p62 is another classical marker to monitor autophagic activity because p62 is selectively degraded by autophagy[32]. In PCT cells, TM treatment greatly increased the amount of accumulated LC3II and conversely induced a drop in p62 (Fig. 5a–f). Remarkably, both TMEM33 and PC2 knock-down prevented TM-dependent LC3II accumulation (Fig. 5a–d). These observations suggest that in renal epithelial cells TMEM33/PC2 links ER

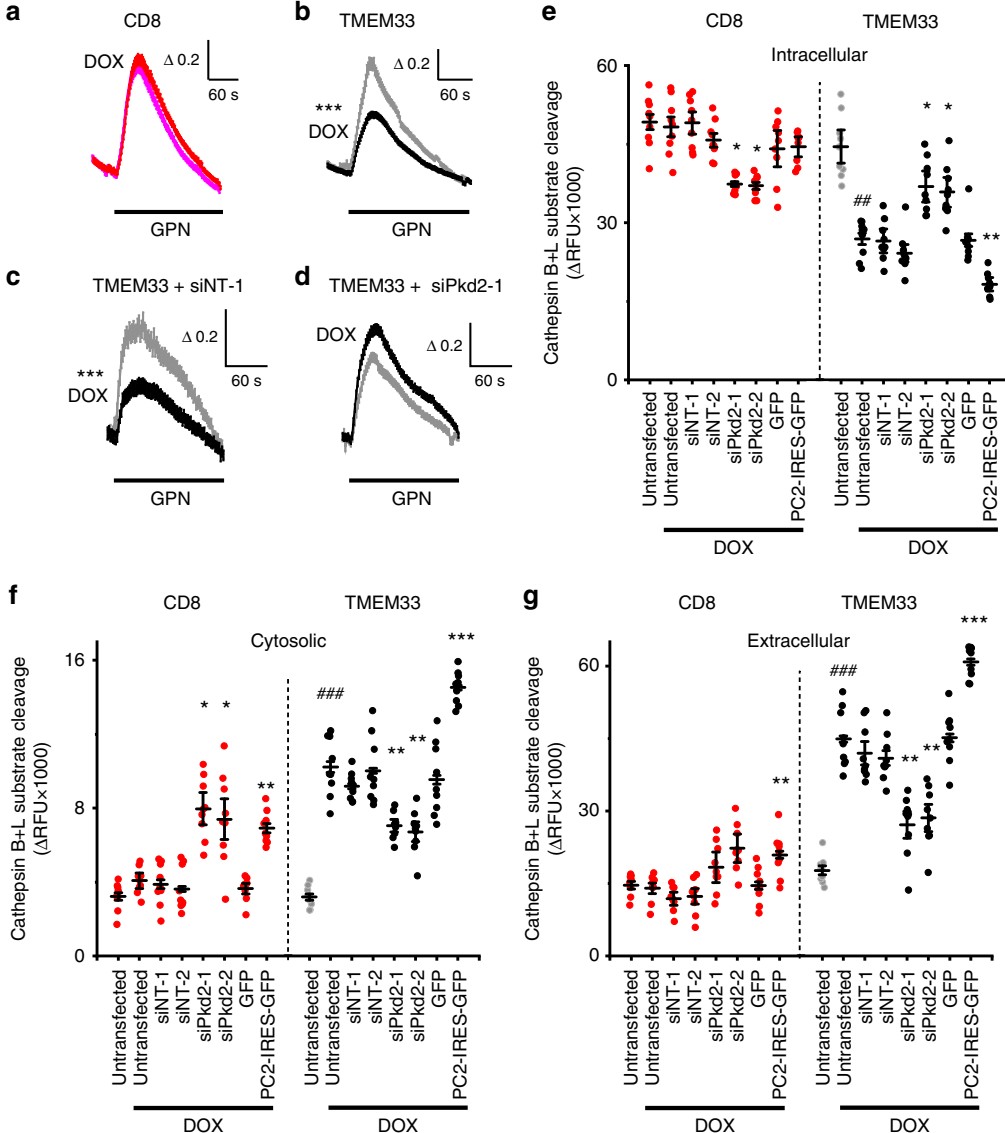

**Fig. 4** TMEM33 inhibits endolysosomal calcium refilling and induces cathepsins translocation. **a** Increases in cytosolic calcium concentration are expressed as ratios of 340:380 nm fluorescence signals ($\Delta R/R_0$). GPN (250 μM) response in the absence of extracellular calcium in a conditional PCT cell line expressing CD8ER in the presence of DOX (magenta trace, $n = 692$), as compared to the non-induced condition (red trace; $n = 706$). **b** Same in a conditional PCT cell line expressing TMEM33 in the presence of DOX (black trace; $n = 548$) or without induction (gray trace; $n = 620$). **c** GPN response in cells transfected with siNT ($n = 204$ and $n = 224$, in the absence [gray trace] or the presence of DOX [black trace], respectively). **d** Knockdown of *Pkd2* (coding for PC2) suppresses the effect of TMEM33 ($n = 361$ and $n = 422$ in the absence and in the presence of DOX, respectively). **e** Effect of *Pkd2* knockdown or PC2 overexpression on intracellular cathepsins B/L content. CD8: red dots; TMEM33 non induced: gray dots; TMEM33 induced: black dots. **f** Same for cytosolic cathepsins. **g** Same for extracellular cathepsins. Values represent average peak $\Delta$RFU values for $n = 3$ independent transfection experiments done in triplicate. * compared to respective untransfected DOX induced controls and # compared to respective non-induced controls. Bars are means $\Delta$RFU ± SEM overlaid with dot plots for e-g. One star indicates $p < 0.05$, two stars $p < 0.01$ and three stars $p < 0.001$, with a Student's t-test used to evaluate statistical significance. Source data are provided as a Source Data file

stress to the regulation of autophagy. LC3II accumulation induced by TM might be due to either a stimulated autophagic flux and/or to a decreased degradation of autolysosomes[32]. Chloroquine (CQ), a lysosomotropic agent, which prevents autolysosomal degradation, caused an accumulation of LC3II in basal conditions, that was further enhanced by TMEM33 expression (Fig. 5e, f)[32]. However, when TM was present (which by itself elevates LC3II), CQ-induced LC3II accumulation in TMEM33 expressing cells was blunted (1.3 fold versus 2.6 fold in the presence of DMSO; Fig. 5e, f). These findings indicate that TMEM33 stimulates the autophagic flux in basal conditions,

while it attenuates autolysosome degradation during TM-mediated ER stress.

Lysosomal dysfunction and translocation of cathepsins have been shown to initiate cell death[33–35]. Next, we explored whether TMEM33 might influence PCT cell death, as a consequence of altered lysosomal function.

**TMEM33 sensitizes PCT cells to apoptotic cell death.** PCT cells from *TMEM33*$^{-/-}$ mice were transiently complemented using plasmids encoding TMEM33 and/or PC2, along with a fluorescent Cherry reporter (IRES construct). Cell viability was

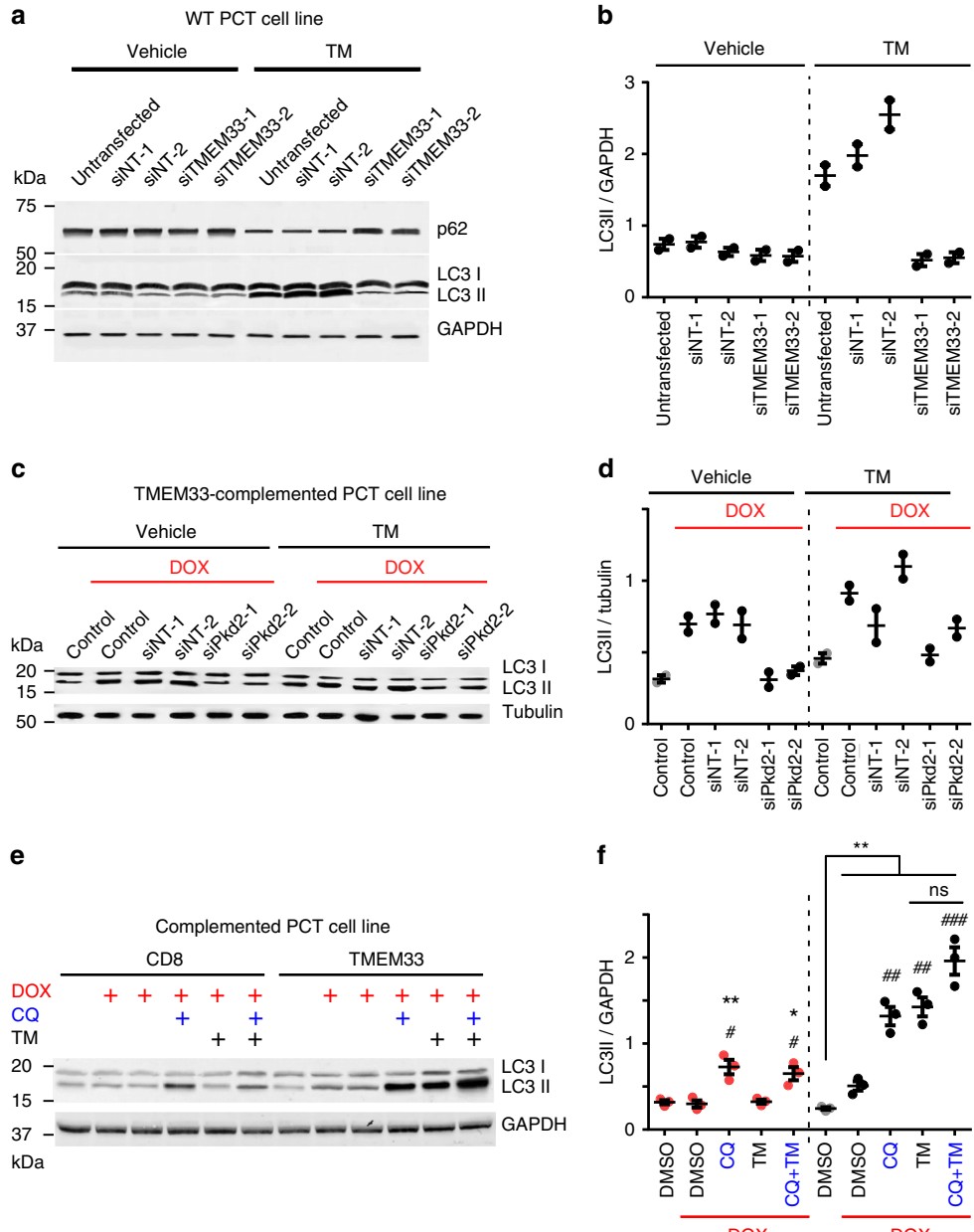

**Fig. 5** TMEM33/PC2 and the regulation of autophagy. **a** p62, LC3II/LC3I and GAPDH expression in WT PCT cells transfected with control siNT or siTMEM33 siRNAs. Cells were either treated with the vehicle or with TM (1 µg/ml for 16 h). Stimulation of autophagy induced by TM is seen as a decreased expression of p62 and increased LC3II/LC3I ratio. **b** LC3II/GAPDH ratio quantified from (**a**). **c** LC3II/LC3I and tubulin expression in TMEM33$^{-/-}$ cells conditionally complemented with TMEM33 (upon DOX induction). Cells were transfected with control siNT or si*Pkd2* siRNAs. Cells were either treated with the vehicle or with TM (1 µg/ml for 12 h). **d** LC3II/tubulin ratio quantified from (**c**). Gray dots: uninduced cells; black dots: induced with DOX. **e** Estimation of autophagic flux in CD8- or TMEM33-complemented TMEM33$^{-/-}$ stably complemented PCT cells. Treatments with doxycyline (DOX) to induce either CD8ER or TMEM33 expression is illustrated in red. Cells are treated either with vehicle (DMSO) or with TM (1 µg/ml) for 8 h with or without chloroquine (20 µg/ml)(CQ) for 2 h. **f** LC3II/GAPDH ratio quantified from **e**. Red dots: CD8; Gray dots: TMEM33 uninduced cells; black dots: TMEM33 induced with DOX. For TMEM33 expressing cells, the difference between DMSO and CQ is significant (**), while the difference between TM and TM + CQ is not significant. Values are means ± SEM overlaid with dot plots for **b**, **d**, **f**. One star indicates $p < 0.05$, two stars $p < 0.01$ and three stars $p < 0.001$, with a two-tailed Student's t-test used to evaluate statistical significance. Source data are provided as a Source Data file

estimated by counting the number of Cherry positive cells using a cell sorter and normalized to the control CD8ER condition (Fig. 6a). TMEM33 significantly lowered cell viability when transfected in a TMEM33$^{-/-}$ background (Fig. 6a). Remarkably, in Pkd2$^{-/-}$ PCT cells this effect was absent, although the effect was recovered when PC2 was also complemented (Fig. 6a). In addition, the number of DAPI positive cells (as an index of membrane permeabilization) was significantly enhanced when

TMEM33 was expressed in TMEM33$^{-/-}$, unlike in Pkd2$^{-/-}$ cells [unless PC2 was complemented] (Fig. 6b). Of note, neither the number of annexin V positive cells nor caspase activity was significantly enhanced by TMEM33 expression in TMEM33$^{-/-}$ cells (Fig. 6c, d).

Next, we used an in vitro model of PCT cells ER stress induced by TM (1 µg/ml; 16 h)[36,37]. In the conditional TMEM33 expressing PCT cell line, DOX addition induced a

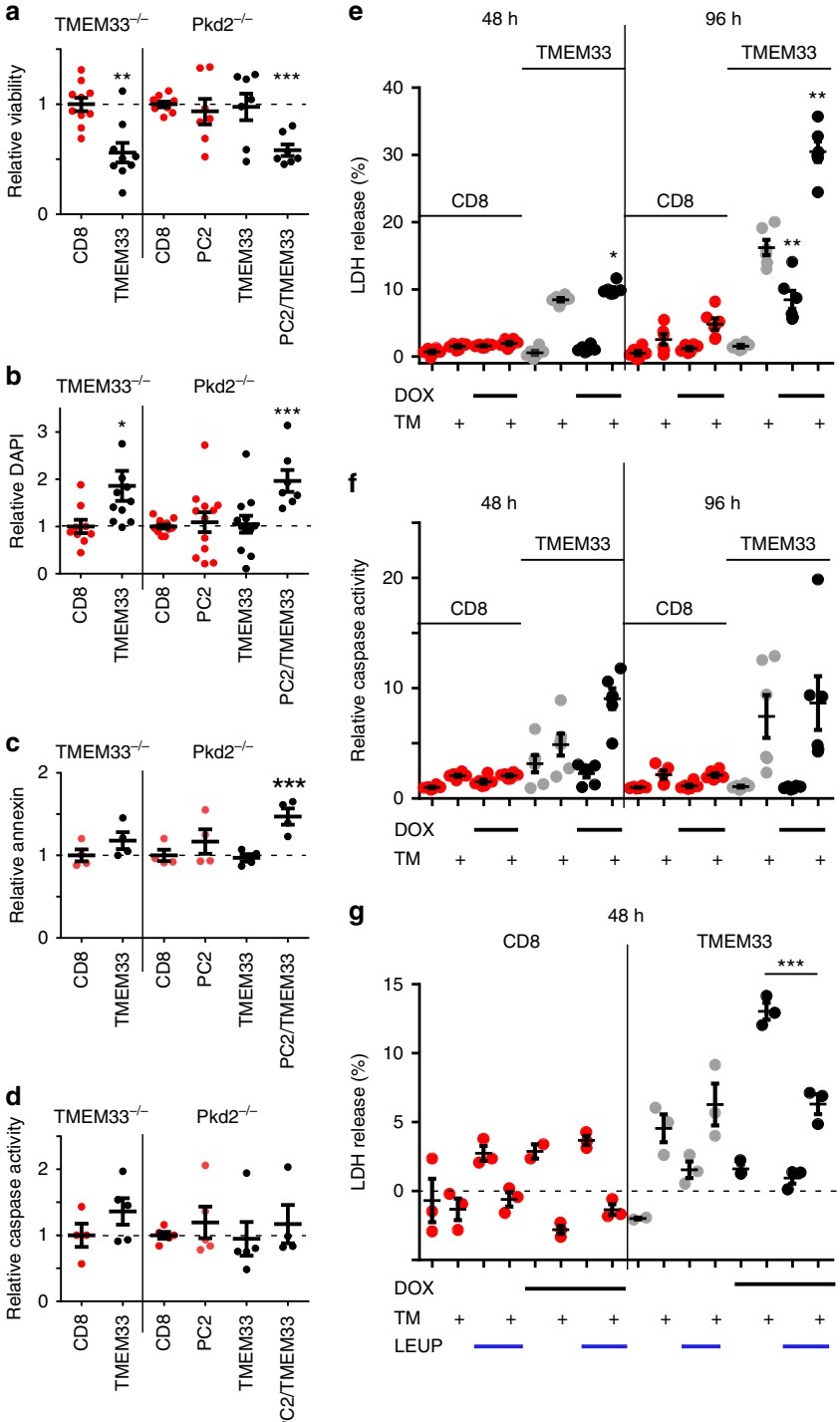

**Fig. 6** TMEM33 overexpression induces cytotoxicity in a PC2-dependent manner. **a** Cell viability determined by measuring the number of Cherry positive cells using a cell sorter. CD8ER (red dots) was used as a negative control. Cells were transfected with TMEM33 (black dots) or PC2 alone (red dots), or with the mix TMEM33/PC2 (black dots). The relative (to the CD8ER conditions) number of viable cells expressing Cherry is illustrated. **b** Relative number of DAPI positive cells. One TMEM33 complemented TMEM33$^{-/-}$ data point (4.66) is out of scale. **c** Relative number of annexin positive cells. **d** Relative caspase 3/7 activity. **e** LDH release in either a conditional PCT cell line expressing CD8ER (in red) or TMEM33 (in black), without (in gray) or with DOX induction (in black) at 48 and 96 h. Cells were treated with vehicle (DMSO) or TM (1 μg/ml) for 16 h. **f** Same for caspase 3/7 activity. **g** Effect of the cathepsins inhibitor leupeptin (LEUP; 25 μM) on LDH release in a conditional PCT cell line expressing TMEM33 (black dots) or CD8ER (red dots) without or with DOX induction (48 h), as indicated. Values are means ± SEM overlaid with dot plots. One star indicates $p < 0.05$, two stars $p < 0.01$ and three stars $p < 0.001$, with a one-way permutation test used to evaluate statistical significance. Source data are provided as a Source Data file

time-dependent (over 96 h) increase in basal and TM-induced release of lactate dehydrogenase (LDH; an index of cytotoxicity), unlike in CD8ER expressing cells (Fig. 6e). LDH release from the TMEM33 expressing cell line in the absence of DOX induction is likely to be related to the leakiness of the repressor system and to the basal TMEM33 expression (TMEM33/TOP1: 0.06 + 0.002) occurring even in the absence of DOX, corresponding to about 10% of the cortical level of TMEM33 (Supplementary Fig. 9a). Caspase 3/7 activity was not significantly influenced by basal TMEM33 expression, although it was strongly enhanced by TM treatment in TMEM33 expressing cells (Fig. 6f). Finally, cathepsins inhibition by leupeptin (LEUP) significantly protected PCT cells expressing TMEM33 from TM-induced LDH leakage, indicating lysosomal dysfunction and contribution of cathepsins (Fig. 6g).

Altogether, these findings indicate that TMEM33 expression causes cytotoxicity (as detected by an increase in LDH leakage and DAPI staining, as well as decreased cell viability) of PCT cells and markedly enhances TM-induced apoptosis. Importantly, the cytotoxic effect of TMEM33 was critically dependent on the presence of PC2.

Next, we investigated in vivo whether TMEM33 might similarly exert a deleterious effect on renal tubular cells in conditions of AKI.

**TMEM33 deletion confers renal protection against AKI.** We took advantage of a constitutive TMEM33 knockout (KO) mouse model and focused on the renal function of TMEM33,

specifically investigating sensitivity to AKI. Homozygote mice were viable, breeding and basal physiological parameters were similar to wild type (WT) mice (Supplementary Tables 1 and 2). In aged mice, over 1 year old, we saw no evidence for morphological anomalies, including the presence of renal cysts (Supplementary Fig. 8). Expression of various ER stress related genes in cortical tissues, including GRP78 and CHOP were unchanged in the KO model (Supplementary Fig. 10). Moreover, renal expression of PC2 was not altered in the KO mice (Supplementary Fig. 9e). Mice were subjected to AKI using the in vivo TM toxicity model[37,38]. Following IP injection of 2 mg/kg TM, about 20% of the WT mice died within 3 days (Fig. 7a, top panel). However, all KO mice survived, despite the fact that the loss in body weight induced by TM injection was identical between both mouse lines (Fig. 7a, top and bottom panels). Of note, the level of TMEM33 expression that is about twice as much in mouse cortex, as compared to the medulla was unaltered in the model of TM-induced AKI (Supplementary Fig. 9a, b). Moreover, the pattern of TMEM33 expression (as visualized by the LacZ reporter) in proximal or distal tubular segments was not visibly altered upon TM treatment (Supplementary Fig. 9c, d). In addition, expression of *Pkd2* or TRPV4 (a partner of PC2 at the primary cilium[39]), was unchanged in the KO without or withTM treatment (Supplementary Fig. 2c, d). Expression of GRP78 and CHOP in the cortex was monitored at 12 and 72 h post TM injection (Supplementary Fig. 10). Both genes peaked at 12 h post injection and subsequently declined at 72 h. We observed no significant difference in the expression of either ER

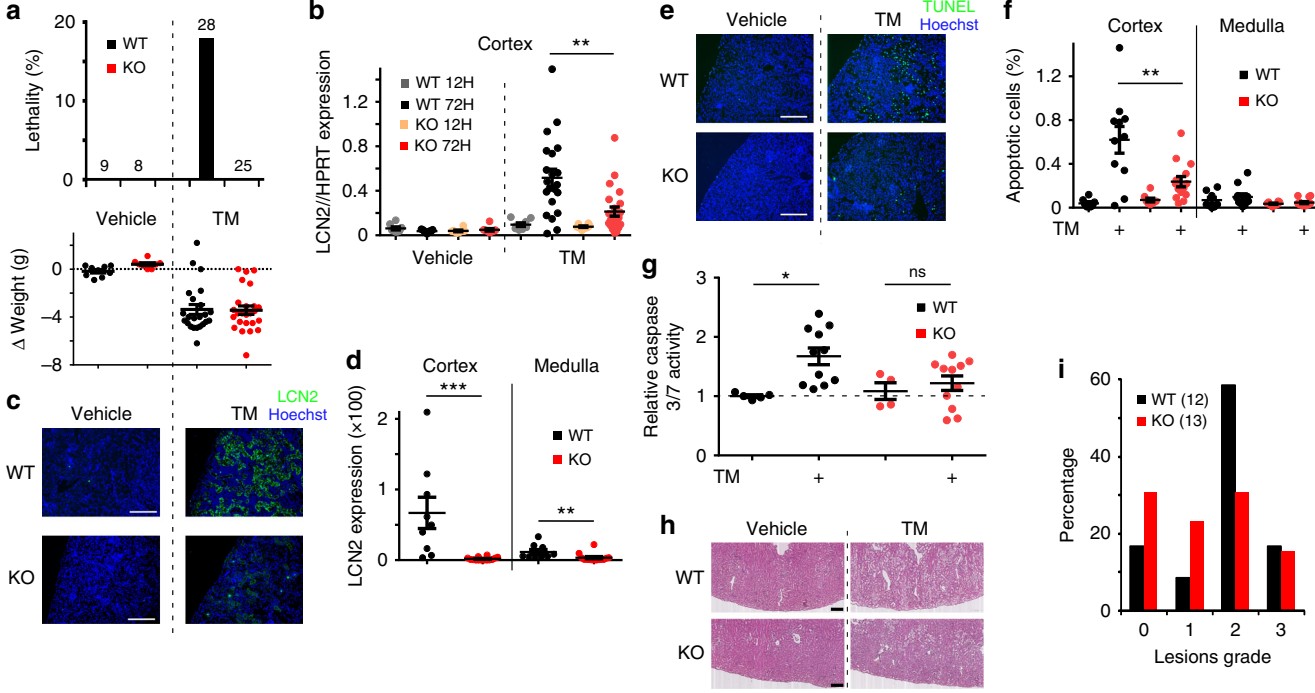

**Fig. 7** Deletion of TMEM33 in the mouse confers protection against AKI. **a** Percentage of lethality (the number of mice is indicated) is shown in the top panel. The bottom panel shows the amount of weight loss induced by TM injection. WT: black bars and dots; KO: red bars and dots. **b** Expression of LCN2, as detected by qPCR in mouse renal cortex 12or 72 h after TM (2 mg/kg) injection, comparing WT (gray and black dots) and KO (orange and red dots) mice. **c** LCN2 and Hoechst staining on cortical sections of kidney from WT and KO mice injected with vehicle or TM (same *n* values as a). Scale bars indicate 200 µm. **d** Relative intensity (TM/vehicle) of LCN2 staining in the cortex and medulla of WT and KO mice. **e** Tunel and Hoechst staining on cortical sections from kidneys of mice injected with vehicle or TM, comparing WT and KO mice. Scale bars indicate 200 µm. **f** Percentage of apoptotic cells in the cortex and the medulla. **g** Relative (to the vehicle treated WT mice) caspase 3/7 activity determined in renal cortical homogenates from WT and KO mice either injected with vehicle or TM. **h** Hematoxylin staining of cortical renal sections of WT and KO mice injected with vehicle or TM. **i** Quantification of lesion grades on cortical sections from WT and KO mice injected with vehicle or TM. The number of mice is indicated on the graph. Values are means ± SEM overlaid with dot plots. One star indicates *p* < 0.05, two stars *p* < 0.01 and three stars *p* < 0.001, with a one-way permutation test used to evaluate statistical significance. Source data are provided as a Source Data file

stress genes between WT and KO mice injected with TM (Supplementary Fig. 10). NGAL/LCN2 (neutrophil gelatinase-associated lipocalin) is routinely used clinically as an early biomarker for AKI[40]. We observed that LCN2 mRNA and protein expression at 72 h post TM injection was significantly decreased in the cortex of KO mice (Fig. 7b–d). A similar effect was found in the medulla, despite major differences in LCN2 mRNA and protein expression levels (about 10-fold higher mRNA level, but 5-fold less protein in the medulla), as seen with two separate antibodies (Fig. 7d and Supplementary Fig. 9f). Next, we estimated the number of apoptotic cells on renal sections using TUNEL staining (Fig. 7e, f). There was no significant difference in basal apoptosis when comparing vehicle injected WT and KO mice. However, in TM-injected mice the cortical increase in apoptotic cell death, as well as caspase 3/7 activation, was blunted in kidneys from KO mice (Fig. 7e–g). These findings were confirmed using Periodic acid-Schiff (PAS) staining and morphological analysis of kidney sections (Fig. 7h, i). No evidence of renal cysts was found with ($n = 12$ and 13, for WT and KO, respectively) or without TM treatment ($n = 4$ and 2 for WT and KO, respectively). Grade 2 cortical lesions in TM-injected mice, characterized by massive tubular dilatation, were reduced by about half in the KO mice (Fig. 7h, i). Thus, a significant protection against TM-induced AKI is observed in the TMEM33 KO mice. In summary, our findings indicate that TMEM33 plays an important role in renal tubular cell vulnerability associated with AKI.

**Does TMEM33 exert a protective effect against cystogenesis?**
It is now well established that reduced PC1/PC2 function (i.e. dosage) causes ADPKD[1]. However, the relative contribution of PC2 at the primary cilium and/or at the ER in the pathology remains unclear at this stage. An important question is whether or not TMEM33 (a selective activator of PC2 at the ER, but not at the primary cilium) might exert a protective effect against *Pkd2*-dependent cystogenesis. In other words, does TMEM33 invalidation aggravate cyst formation caused by *Pkd2* knock-down and can we probe the functional role of ER PC2 in renal cystogenesis taking advantage of its activator TMEM33? It is important to note that a protective effect of TMEM33 would not be expected if *Pkd2* is fully invalidated (homozygote knockout), since TMEM33 could not enhance PC2 channel opening (Fig. 2e, f). Exploring these questions in mice is technically challenging as a *Pkd2* hypomorphic model together with a targeted deletion of *TMEM33* would need to be implemented (of note *Pkd2*$^{+/-}$ mice are not cystic and constitutive *Pkd2*$^{-/-}$ is embryonic lethal[41]). To circumvent this major difficulty, we instead turned to a more amenable and validated zebrafish model of renal cystogenesis based on the injection of morpholino oligomers to knock down *pkd2* and recapitulate the molecular mechanism (dosage effect) of ADPKD and induce renal cysts[42]. The zebrafish pronephros is composed of two nephrons with glomeruli fused at the midline (Fig. 8a). In the zebrafish ADPKD model, a translation blocking morpholino oligomer complementary to the ATG of *pkd2* (MO *pkd2*) is injected into the embryos at the 1 cell stage, causing a large dilation of the glomeruli (cysts) two days later[42] (Fig. 8b). Taking advantage of this powerful and accessible assay, we compared the effect of pkd2 *knockdown* on glomerular enlargement (using the *wt1b:EGFP* fluorescent marker) in WT zebrafish (*tmem33*$^{+/+}$) and in a *tmem33* mutant fish line[43]. Glomerular size of *tmem33*$^{+/+}$, *tmem33*$^{+/-}$ or *tmem33*$^{-/-}$ embryos injected with control MO was identical (Fig. 8c). Importantly, enlargement of glomeruli (i.e. cysts) induced by pkd2 MO was not significantly different in *tmem33*$^{+/-}$ or *tmem33*$^{-/-}$ embryos, as compared to WT embryos (Fig. 8c).

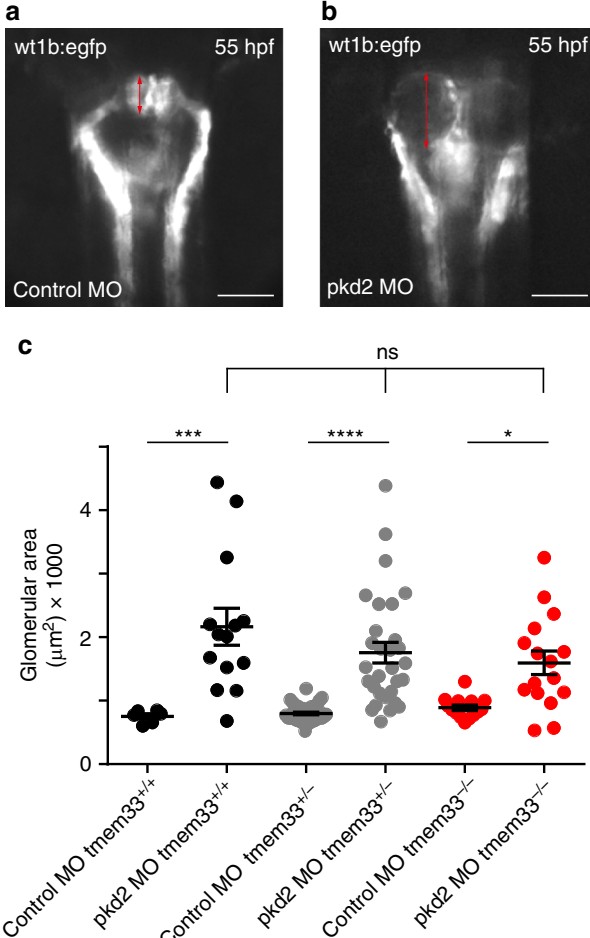

**Fig. 8** Tmem33 does not influence renal cystogenesis in zebrafish. **a** Epifluorescence image of zebrafish glomeruli. Control morpholino oligomer injected in *Tg(wt1b:EGFP)*$^{li1}$. **b** Morpholino knockdown of *pkd2* increases glomerular area in *Tg(wt1b:EGFP)*$^{li1}$ (red arrows). Scale bars: 50 μm. **c** *Pkd2*-dependent renal cystogenesis was unaffected by the loss of tmem33 in zebrafish *tmem33*$^{sh443}$ mutants (tmem33$^{+/+}$ black dots, tmem33$^{+/-}$ gray dots and tmem33$^{-/-}$ red dots). Each data point refers to a single glomerulus. Values are means ± SEM overlaid with dot plots. One way ANOVA with Turkey's post hoc *$p < 0.05$, ***$p < 0.001$, ****$p < 0.0001$, F = 15.44, DF = 119, 2 repeats. Source data are provided as a Source Data file

**Discussion**

In this report, we demonstrate that TMEM33 is found in a complex together with the ER channel PC2 in PCT cells. Deletion of TMEM33 enhances IP$_3$-dependent calcium signaling, whereas its overexpression has the opposite effect. Strikingly, these effects on the regulation of intracellular calcium homeostasis critically depend on PC2. TMEM33 promotes PC2 channel opening over the whole cytosolic calcium range in lipid bilayer ER liposomes reconstitution experiments. Inhibition of IP$_3$ signaling by TMEM33 lowers calcium refilling of endolysosomes, causing lysosomal enlargement, cathepsins translocation and impaired autophagic flux during ER stress. Moreover, TMEM33 enhances TM-induced PCT cytotoxicity in vitro that is prevented by cathepsins pharmacological inhibition. In line with these findings, genetic deletion of TMEM33 exerts a significant protection in a mouse model of TM-induced ER stress. Thus, TMEM33 influences the regulation of intracellular calcium homeostasis in kidney tubular epithelial cells and is associated with enhanced

susceptibility to AKI, while it does not affect *pkd2*-dependent renal cystogenesis.

TMEM33 is a transmembrane protein that is conserved during evolution. There are two isoforms in the budding yeast *S. cerevisiae* called Pom33 and Per33[16]. Pom33 is present at the ER and is also dynamically associated with the nuclear pore complex, unlike Per33 that is restricted to the ER[16]. Pom33 contributes to the distribution and/or assembly of nuclear pores[16,44]. The fission yeast TMEM33 protein (called Tts1) also functions in organizing peripheral ER and in remodeling the nuclear envelope during mitosis[17,18]. It is predicted that TMEM33 is made of one transmembrane segment followed by a hairpin in which hydrophobic segments do not fully span the membrane, with both N and C termini facing the cytosol[16]. The C-terminal amphipathic helix in Tts1 plays a key role in ER shaping and modulating the mitotic nuclear pore complex distribution[17]. In addition, Tts1/ Pom33 might function to sustain the highly curved ER domains in interaction with the reticulons Rtn1 and Yop1[16–18]. In mammalian cells, TMEM33 was similarly characterized at the ER membrane as a Rtn-binding protein[19]. Interestingly, TMEM33 has the ability to suppress the membrane-shaping activity of Rtns, thereby influencing the ER tubular structure[19]. In addition, TMEM33 was identified as a participant in US2-mediated degradation of MHC I[45]. Moreover, TMEM33 was shown to be a stress-inducible ER protein that modulates the unfolded protein response signaling by interacting with PERK and IRE1α in cancer cells[46].

The in vivo physiopathological function of mammalian TMEM33 remained to be determined. Here, we demonstrate that TMEM33 is a key modulator of intracellular calcium homeostasis through the regulation of PC2 at the ER, influencing AKI in the mouse. Unexpectedly, we found that TMEM33 expression dramatically alters the calcium-dependency of PC2 in ER liposomes fused to planar bilayer. Increasing cytosolic calcium typically produces a bell-shaped increase in the open channel probability of PC2, indicating the overlapping influence of both a stimulatory (at the lower calcium range) and an inhibitory calcium-dependent mechanism (at the higher calcium range)[7,10]. When TMEM33 is overexpressed, the inhibitory mechanism disappears, thus resulting in a net stimulation of channel activity over the entire physiological calcium range. Our data are consistent with the fact that enhanced PC2 channel opening in the presence of TMEM33 results in ER calcium depletion and diminished IP3 responses, in line with a previously reported role for PC2 as a leak calcium conductance of the ER[14]. Thus, our in vitro experiments demonstrate that TMEM33 is a major regulator of ER PC2 gating (i.e. acting as an auxiliary subunit), promoting its opening over the whole cytosolic calcium range. Our findings further validate PC2 as an ER calcium release channel[7,12,14].

Acidic organelles are coupled to the ER through calcium microdomains at membrane contact sites. ER calcium content and IP3R-evoked calcium release are proposed to control the uptake of calcium in lysosomes[26,27,47]. We observed that endolysosomal calcium load is reduced by TMEM33, along with diminished IP3 responses. Endolysosomal size and translocation of cathepsins is enhanced by TMEM33. We propose that altered lysosomal calcium refilling through impaired IP3-dependent signaling is a major contributor to these anomalies. Previous findings indicated that larger lysosomes are more susceptible to breakage and release of their contents[48]. Thus, lysosomal leakiness induced by TMEM33 might possibly be related to the increase in their size.

The effect of TMEM33 on the regulation of intracellular calcium homeostasis shown in vitro is critically dependent on PC2. Similarly, TMEM33-induced translocation of cathepsins was greatly influenced by PC2. In addition, the cytotoxic effect of

TMEM33 was again dependent on PC2. However, we cannot rule out that some of the TMEM33 effects might be PC2-independent. TMEM33 was previously linked in various cell types to different mechanisms, including tubulation of the ER, nuclear pore assembly and distribution, protein degradation or ER stress responses (see above). Thus, it is possible that these mechanisms might also be at play in the kidney. For instance, previous findings demonstrated that TMEM33 interacts with Rtns in both yeast and mammalian cells, influencing ER tube caliber[16–19]. Whether or not TMEM33 might influence PC2 and/or IP3R activity through a change in local ER membrane curvature is a possibility to be considered. Moreover, changes in membrane curvature may also impact endolysosomal morphology and ability to form membrane contact sites at the ER. Notably, ER/endosome contact also regulates maturation and fission of endosomes[49,50]. Thus, disruption of ER/lysosome contact may be an additional explanation for depletion of lysosomal calcium content. In this context, TMEM33 might also act as a negative regulator of membrane contact sites between the ER and the endolysosomes.

In the human breast cancer cell line MCF-7, TMEM33 interacted and stimulated both the PERK/p-eIF2α/ATF4/CHOP and IRE1α-XBP1-S signaling pathways[46]. However, the present findings indicate that TMEM33 sensitizes PCT cells to TM-induced apoptosis, independently of a change in the ER stress response. Instead, our study indicates that kidney lysosomal dysfunction, including translocation of cathepsins contribute to the cytotoxic effects of TMEM33. Thus, renal tubular epithelial cells and breast cancer cells show a different susceptibility to the deleterious effects of TMEM33 expression, with the possible differential involvement of cell cycle components or other elements that contribute to cancer[46].

Clearly, future studies will be required to identify all the mechanisms at play (including those independent of PC2) in the renal effects of TMEM33. Nevertheless, the present work already brings strong evidence that TMEM33 might behave as an auxiliary regulatory subunit of ER PC2 in PCT cells, acting as a gating activator (at least demonstrated in vitro in the present report). In mammary breast cancer cells, TMEM33 mRNA and protein expression are greatly enhanced by thapsigargin or TM treatments[46]. In the same line, it is interesting to note that during ischemic acute renal failure, affecting primarily the proximal tubule, a pronounced upregulation of intracellular PC2 (the partner of TMEM33) was reported[22–24]. The upregulation of PC2 might act as a brake on cell proliferation (acting in part as a tumour suppressor gene) to allow the proliferative index of the ischemic kidney to return to baseline[23]. Accordingly, the magnitude and duration of tubular and interstitial proliferative responses was enhanced in injured *Pkd2*$^{+/-}$ mice[51].

Dysregulation of cathepsins expression/activity is associated to the onset and progression of various kidney diseases, including AKI[34]. For instance, cathepsin D is increasingly recognized as a key driver of apoptosis during AKI[52]. In addition, translocation of cathepsins from the lysosome into the cytoplasm was demonstrated as an important event involved in renal physiopathology[53]. Thus, the previously reported multifaceted role of cathepsins in kidney disease is clearly in line with the present study[34].

Autophagy is triggered upon stress to eliminate excess proteins and protects cells against metabolic damage[54–58]. It has been suggested that increased $[Ca^{2+}]_{cyt}/[Ca^{2+}]_{ER}$, at least partly, contributes to the stimulation of autophagy[59]. Recent findings indicate that in cardiomyocytes, PC2 functions to promote autophagy under glucose starvation or mTOR (target of rapamycin) inhibition[29,31]. Knockdown or knockout of PC2 reduced the autophagic flux, while

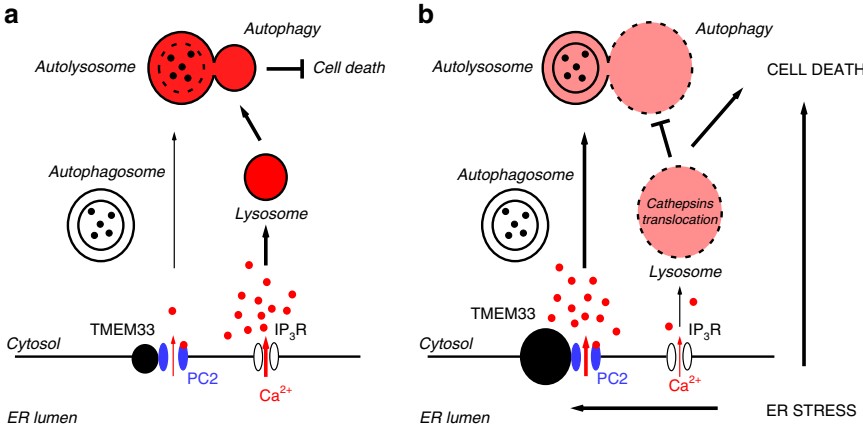

**Fig. 9** TMEM33/PC2, intracellular calcium homeostasis and cell death. Schematic model describing the effect of TMEM33 on the regulation of intracellular calcium homeostasis, lysosomal function, autophagic flux and tubular cell death in control conditions (**a**) or upon ER stress (**b**). The proposed model (figure created by EH) addresses the scenario of AKI (**b**), when TMEM33 expression is enhanced[46] (right panel). Our findings indicate an interaction between TMEM33 and PC2 at the ER membrane causing an increase in PC2 channel activity spanning the whole physiological calcium range. TMEM33 induces, through PC2, a decrease in intracellular calcium and a diminished $IP_3$ calcium signaling (right panel). Consequent decrease in the calcium refilling of lysosomes, associated with an enlargement and translocation of cathepsins sensitize PCT cells to TM-induced apoptosis. Moreover, impairment of autolysosome degradation causes an inhibition of the autophagic flux and a loss of cellular protection upon ER stress. Accordingly, genetic deletion of TMEM33 in the mouse provides a significant protection against TM-induced AKI. Source data are provided as a Source Data file

PC2 overexpression had the opposite effect[29,31]. PC2-induced autophagy in cardiac cells was blunted by intracellular calcium chelation, whereas removal of extracellular calcium had no effect[29]. These findings suggest a model whereby PC2-dependent regulation of autophagy occurs through the regulation of intracellular calcium homeostasis[29,31] (Fig. 9a). Previous findings also indicated that TMEM33 overexpression in cancer cells results in a stimulation of autophagy[46]. Similarly, in the present study we observed an enhancement of the autophagic flux of PCT cells by TMEM33 in the basal conditions (Fig. 5a–f). Activation of ER PC2 by TMEM33 at least partly contributes to this effect (Fig. 5c and Fig. 9a). By contrast, upon TM-induced ER stress we observed an impaired autophagic flux when TMEM33 was expressed, likely due to lysosomal dysfunction (Fig. 5e, f, Fig. 9b).

Altogether, our findings suggest that TMEM33 is an important, so far unrecognized player in AKI. In basal conditions, TMEM33 overexpression does not induce signs of apoptosis (as detected by caspase 3/7 activity or TUNEL staining). Rather, we show that TMEM33 sensitizes PCT cells to apoptosis during ER stress. We propose a mechanistic model addressing the scenario of AKI, including a possible role of TMEM33/PC2 in cathepsins translocation and impaired autophagic flux upon ER stress (Fig. 9b). Accordingly, deletion of TMEM33 exerts a potent protective role against TM toxicity (Fig. 7).

Since TMEM33 acts as an activator of PC2, at least in vitro, we investigated whether or not it might exert some beneficial effects against *Pkd2*-dependent cystogenesis. Using a *pkd2* hypomorphic model (dosage) of ADPKD in the zebrafish, we found no significant protection of TMEM33 against *pkd2*-dependent renal cystogenesis. These findings indicate that ER PC2 might not be causal to ADPKD, but rather a loss-of-function of PC2 at the primary cilium (that does not interact with TMEM33) is responsible for cystogenesis, as already indicated by the tight association between renal cystic diseases and numerous ciliopathies[60].

In conclusion, our in vitro work shows that TMEM33 acts as a gating activator of ER PC2. Moreover, our in vivo studies suggest that TMEM33 is involved in AKI, while it does not influence renal cystogenesis, at least in the zebrafish.

## Methods

**Cell culture.** PCT cells were derived from mouse kidneys and primary cultures were immortalized with pSV3NEO[61]. Cells were grown in DMEM/F12 (V/V) medium, containing 1% FBS, 15 mM NaHCO₃, 2 mM glutamine, 20 mM HEPES, 5 mg/l insulin, 50 nM dexamethasone, 10 μg/l EGF, 5 mg/l transferrin, 30 nM sodium selenite, 10 nM T3 (triiodo-L-thyronine) and 125 μg/ml G418 (geneticin).

PCT *Pkd2⁻/⁻* cells[62] (a kind gift from Steve Somlo) were grown in DMEM/HAMF12 (V/V) medium, containing 3% FBS, 7.5 nM Na selenite, 1.9 nM T3 (triiodo-L-thyronine), 5 mg/l insulin, 5 mg/l transferrin, 50 U/ml nystatin, 10 U/ml INF (interferon-gamma) and 100 μg/ml penicillin/streptomycin at 33 °C in a humidity-controlled incubator with 5% CO₂. HeLa cells (a kind gift from Valérie Doye;[16]) were cultured in DMEM containing 10% FBS and 100 μg/ml penicillin/streptomycin. For HeLa stably expressing GFP-TMEM33 1 mg/ml of G418 was added.

**Plasmid constructs.** The following plasmids were used in this study: pIRES2-EGFP PC2 (#32), pIRES2-EGFP TMEM33 (#212), pBUD mCherry-PC2 (#250), pBUD 1.mCherryPC2 2.EGFPTMEM33 (#255), pIRES-puro3 HA-PC2 (#203), pBUD 1.TMEM33iresCherry 2.PC2 (#438), pBUD 1.CherryiresCherry 2.PC2 (#439), pBUD 1.TM33iresCherry 2.CD8ER (#440), pBUD 1.CherryiresCherry 2. CD8ER (#441). pCDNA3.1 CD8-ER (I-II loop Ca_v a 1 A)-Myc (#437). pCMV Myc-PC2 (#113), pCMV Myc-742X (#116), pcDNA3.1 Myc-TMEM33 (#217), pcDNA3.1 HA-PC2 (#179), pcDNA3.1 HA-742X (#180), pcDNA3.1 TMEM33-HA (#177), pCDNA3,1 zeo (+)mCherry -PC2 (# 253), pEGFP-N1 -TMEM33 (#106). Samples and maps are available on request.

**TMEM33 knockout mice.** ES cells heterozygous for TMEM33 (Tmem33 tm1 (KOMP)Mbp) were purchased from the KOMP (Knockout Mouse Project) repository UC Davis, CA, USA. The construct contains a synthetic cassette including LacZ, inserted in intron 2–3 and deleting coding exons 3, 4, 5 and 6 (Supplementary Fig. 11). The remaining exons encode 27 amino acids of which 15 correspond to the predicted cytosolic amino terminus of TMEM33. TMEM33 knockout mice were generated using these ES cells at MRC Harwell, Oxford, UK.

**Generation of stable cell lines using the T-Rex system.** As a first step, immortalized PCT cells derived from (TMEM33⁻/⁻) KO mice were transfected with pcDNA6TR (Tet repressor) linearized with FspI. Transfections were performed in a 35 mm plate with 250 000 cells using 2 μg of plasmid and 5 μl of Lipo2000 (Invitrogen) according to manufacturers instructions. After 48 h, cells were dissociated with 250 μl trypsin and 750 μl of complete medium. The serum used was TET system approved FBS (Clontech). 25 μl, 50 μl and 100 μl were then seeded into 10 cm dishes with medium supplemented with 15 μg/ml blasticidin. Individual clones were picked using cloning cylinders and amplified in a 6-well plate. Each clone was tested by western blot to determine the best expression of the Tet repressor using the Tetr antibody from Sigma. Clones were further tested using the pcDNA4/TO lacz plasmid +/− 1 μg/ml of DOX. Clone 34 was selected to introduce pcDNA4/TO TMEM33iresCherry (#368). Transfections were carried out

with plasmid linearized with FspI. 48 h later 25 μl, 50 μl and 100 μl of dissociated cells were seeded into 10 cm dishes with medium supplemented with 25 μg/ml zeocin. A total of 24 clones were analyzed by qPCR after 48 h of induction. Despite the presence of the Tet repressor, all the clones showed some leakiness in the absence of DOX. Cl.34 R (Tetr) was selected to generate a control line expressing pcDNA4/TO CD8-ER iresCherry. Similarly, cell lines were generated to express either pcDNA4/TO TMEM33-HA (#367) and pcDNA4/TO HA-TMEM33 (#366).

**Cell transfection**. Transfections were performed using either JetPEI (Polyplus-transfection) with a DNA:Lipid ratio of 1:2 or Fugene6 (Promega) with a ratio of 1:4 or Viafect (Promega) with a ratio of 1:4 or Lipo2000 with a ratio of 1:2.5. Cells were analyzed 48–96 h later.

Ready-to-use si*TMEM33* and si*Pkd2* (20 nM siRNA) were transiently transfected into different PCT cell lines, using the HiPerFect Transfection Reagent (Qiagen SA, Courtaboeuf, France). Control siRNA experiments were performed by transfecting non-Targeting siRNAs. Cells transfected with siRNA were used 48 h later for mRNA extraction and 72 h for protein extraction and functional analysis. Control siRNA were purchased from Dharmacon (siNT-1) and Invitrogen (siNT-2), siRNA directed against *TMEM33* were purchased from Invitrogen (si*TMEM33–1*: 5′-GGCUGCAUCAGAGAUUACCUCACUU-3′) (si*TMEM33–2*: 5′-GGCUUUCUCGCCCUGUUCACAGUUUA-3′). siRNA directed against *Pkd2* were purchased from Dharmacon (si*PKD2–1* 5′-UCAUAGACUUCUCGGUGUA-3′) (si*PKD2–2* 5′-UACGGGAGCUGGUCACUUA-3′). Validation of the siNRAs against *TMEM33* is shown in Supplementary Fig. 2a-b. Transfection rates of WT (45%, $n = 3$) and TMEM33$^{-/-}$ (41%, $n = 3$) PCT cell lines were determined at 48 h post transfection using siGLO transfected with RNAiMax using Cytation 5 Cell Imaging Multi-Mode Reader.

**Yeast two-hybrid system**. The yeast two-hybrid system used allows the identification of molecular interactions within the nucleus (including cytosolic fragments). Transformations were carried out according to the manufacturer's instructions (Mo Bi Tech). As a first step, the pEG202 series of plasmids were transformed individually into the EGY48 strain and plated onto glucose – HIS. Colonies were then streaked onto glucose –HIS –LEU (to check for autoactivation). Once autoactivation had been ruled out, the pJG4–5 series of plasmids were transformed as shown in Supplementary Table 3.

Fragments cloned into pEG202 or pJG4–5 are as follows: TMEM33 N term (amino acids 1–24), TMEM33 C term (amino acids 197–248), PC2 N term (amino acids 1–222) and PC2 C term (amino acids 677–968)

These transformations were plated onto glucose – HIS– TRP subsequently streaked onto gal/raf –HIS–TRP–LEU. Clones that grew on these plates were then verified for galactose dependence by a drop test. Briefly, yeast were inoculated into 35 μL of 0.9% NaCl. Either 3 μL was spotted onto glucose–HIS–TRP–LEU or gal/raf –HIS–TRP–LEU, a 10-fold dilution was similarly spotted. Real positives were selected on the basis of growth on galactose/raffinose. The prey (pEG202) and bait (pJG4–5) vectors, as well as the yeast (EGY48 strain) were purchased from MoBiTech. The YPDA medium and the DropOut (DO) supplemented mediums were purchased from Clontech.

**qPCR**. Total RNA was isolated using RNeasy mini kit (Qiagen) and equal amounts of cDNA were synthesized using the SCIII reverse transcriptase (Invitrogen). Primer sequences used for quantitative PCR (qPCR) analysis are listed in Supplementary Table 4.

**Western blots**. PVDF membranes (Perkin Elmer) were saturated in PBS 0.1% Tween and 5% non-fat milk then incubated overnight with diluted primary antibodies. After 3 × 15 min washes in PBS 0.1% Tween, secondary antibodies conjugated to HRP were added at a dilution of 1:30,000 (Jackson) and incubated for a minimum time of 1 h. After several washes, the membranes were processed for chemiluminescent detection using the Super Signal West Pico chemiluminescent substrate (Pierce), according to the manufacturer's instructions for detection of Calnexin or GAPDH. Western Lightning ECL Ultra (Perkin Elmer) was used when signals could not be detected using the pico substrate. The membranes were then exposed using the Fusion FX (Vilber Lourmat). The intensity of the signals was evaluated by densitometry and semi-quantified as the intensity of band corresponding to the protein of interest divided by the intensity of the band corresponding to GAPDH, tubulin, actin or calnexin for each experiment. Each experiment presented was repeated at least twice. Uncropped versions of all blots are available within the Source Data file.

**Immunoprecipitations**. Cells were seeded at 500,000 in 60 mm plates the day before transfection. Transfections were performed in duplicate with 2.5 μg of each plasmid using either JetPEI or Lipo2000. 2 plates per transfection were harvested 48–72 h later in 1 ml NEB lysis buffer (150 mM NaCl, 10 mMTris.Cl pH7.4, 1 mM EDTA, 1 mM EGTA, 1% TritonX100, 0.5% NP40 supplemented with Roche Ultra protease inhibitors, PefablocSC, Phosphatase inhibitors). The samples were sonicated and then rotated in the cold room for 30 minutes, before centrifugation at 4 °C for 10 minutes at 20,000 × *g*. A concentration of 20 μg of protein was removed for input analysis. Equivalent quantities of protein were used for the immunoprecipitations. To avoid detecting antibodies during the western blot, 2 μg of antibody (anti-HA 3F10 Roche) was crosslinked to either Protein A or Protein G magnetic beads (NEB) using dimethyl pimelimidate dihydrochloride (D8388 SIGMA). After 30 min preclearing, 25 μl of crosslinked beads were added to each lysate and incubated with rotation in the cold room for a minimum time of 1 h. Beads were washed 3Xs with NEB buffer. Proteins were released from the beads using 25 μl of 0.1 M glycine pH2.5. 5 μl of 1 M Tris.Cl pH7.5 was added to neutralize the acidity. Samples were charged with Laemmli loading buffer/β−mercaptomethanol after heating at 70 °C for 10 min.

**Antibodies**. Antibodies used in this study, as well as dilutions are listed in Supplementary Table 5.

**Lysosomal preparation**. Cells (TMEM33$^{-/-}$, HA-TMEM33 and TMEM33-HA) were seeded in 4 large 175 cm$^2$ flasks and grown to confluence in the presence of doxycycline for 72 h. Cells were trypsinized, spun down and rinsed with 1× PBS- and resuspended in Lysosome Enrichment Reagent A complemented with protease inhibitors. Lysosomes were then prepared according to the manufacturers instructions (ThermoSCIENTIFIC cat. 89839). For western analysis, either 20 μg of whole cell lysate or 2 μg of lysosomal lysates were loaded on a 4–15% gradient gel (Criterion, Biorad). Blots were developed either with Western Lightning Ultra (Perkin Elmer cat. NEL112001EA) or Western Lightning ECL Pro (Perkin Elmer cat. NEL120001EA).

**Intracellular calcium measurements**. Cells were plated, at a density of 3000 cells/cm$^2$, on glass-bottom fluorodishes (World Precision Instruments) coated with collagen. Cells were loaded with 2 μM Fura-2 AM for 30 min. Loaded cells were visualized under an inverted epi-fluorescence microscope (AxioObserver, Carl Zeiss, France) using a Fluar ×20. Then cells were continuously superfused with control or test solutions at 37 °C using an automatic heater controller system (TC-344B temperature/heater controllers, Warner Instrument Corporation, Hamden, USA). The control solution contained (mmol/L): NaCl, 116; KCl, 5.6; CaCl$_2$, 1.8; MgCl$_2$, 1.2; NaHCO$_3$, 5; NaH$_2$PO$_4$, 1; HEPES, 20; glucose 1 g/l, pH 7.35. The excitation light was supplied by a high pressure Xenon-arc lamp (Lambda LS, Sutter Instrument Company, One Digital Drive, Novato, USA) and the 340 and 380 nm wavelengths were selected through a high-speed multi-filter wheel (Lambda 10–3, Sutter Instrument Company, One Digital Drive, Novato, USA). For each excitation wavelength, the fluorescence emission was discriminated by a 400 LP dichroic mirror and a 510/40 bandpass filter. Fluorescence images were collected every 2 seconds by an EMCCD camera (Cascade 512, Roper Scientific, Evry, France). Images were digitized, and integrated in real time by an image processor (Metafluor, Princeton, NJ, USA). 340 and 380 background fluorescence signals were collected at the same rate and subsequently subtracted from respective fluorescent images. Results (ΔR/R0) were expressed as ratios between 340 and 380 fluorescence signals measured during a response divided by the ratio measured in resting conditions (that is, before the addition of an agent). Cells with increased ΔR/R0, in response to different stimuli, were counted from fields containing 30 to 100 cells taken from 3 to 4 separate experiments. First, cells were superfused with the control solution and then CaCl$_2$ was omitted and 1 mM EGTA was added. Agents (ATP 20 μM or GPN 250 μM) were added to the calcium free solution to induce calcium release from the ER or ionomycin (5 μM) to estimate the amount of intracellular calcium.

**Confocal microscopy and immunohistochemistry**. Cells were plated, at a density of 3000 cells/cm$^2$, on glass-bottom fluorodishes (World Precision Instruments) coated with collagen. Cells were transfected 48 h before imaging using Leica SP5 confocal microscope with a 63X objective. Nuclei were labeled with Hoechst staining (blue). A TMEM33-EGFP construct was stably expressed in Hela cells. Primary cilia were visualized with acetylated tubulin antibody (Sigma T7451; 1/200) and a secondary donkey anti-mouse antibody (Alexa 647 Invitrogen A-31571; 1/500). ER was marked with ER Tracker blue (invitrogen E12353) and lysosomes by Lysotracker red (invitrogen L7528).

**Electron microscopy**. Cells were fixed in 1.6% glutaraldehyde in 0.1 M phosphate buffer (pH7.4). They were rinsed with cacodylate buffer 0.1 M, then post-fixed in osmium tetroxide (1% in cacodylate buffer) reduced with potassium ferrycyanide (1%) for 1 h. After a water wash, cells were dehydrated with several incubations in increasing concentrations of ethanol and embedded in epoxy resin (EPON). Eighty nanometer sections were contrasted with uranyl acetate (4% in water) then lead citrate and observed with a Transmission Electron Microscope (JEOL JEM 1400) operating at 100 kV and equipped with a Olympus SIS MORADA camera.

**Gating strategies used for cell sorting**. To ensure high purity of cells, a step-by-step gating strategy was performed (Supplementary Fig. 12). Cells were separated from debris by forward scatter (FSC) and side scatter (SSC) parameters. Using FSC-width (FSC-W) versus FSC-height (FSC-H) and SSC-W versus SSC-H, single cells were gated to avoid false positive signals produced by aggregates. Single cells

were then gated as the mCHERRY + population (living cells) from which DAPI + and caspase + (or annexin + ) were further gated.

**Caspase assay.** A total of 50,000 cells were seeded in a 24-well plate with and without 1 µg/ml DOX. The next day, cells were treated either with vehicle (DMSO) or TM (1 µg/ml). 16 h later, cells were scraped into 50 µl of caspase lysis buffer (1% TritonX100, 150 mM NaCl, 0.1 mM EDTA, 20 mM Hepes pH7.5, 1 mM MgCl$_2$, 5 µg/ml leupeptin, 10 µg/ml aprotinin, 0.1 mM pepstatin A) and incubated on ice for 30 min. Lysates were centrifuged for 5 minutes at 4 °C at 20,000 × g. Supernatants protein dosage was performed using the DC protein assay kit (Bio-rad). 10 µg of protein was used in the assay in a final volume of 100 µl. Caspase-Glo® 3/7 Assay Systems (Promega) was used to quantitate caspase activity.

**Flow cytometry caspase.** Caspase activity was assayed using CellEvent Caspase-3/ 7 Green flow Cytometry assay kit (molecular probes). Cells were seeded in 6-well plates at 250,000 per well the day before transfection. The cells were transfected using 1.5 µg DNA and Lipo2000. After 72 h, supernatants were recuperated. A concentration of 1 ml of accutase (Sigma) was added per well to recuperate the cells. Cells were centrifuged for 5 min at 150 × g and the pellet was resuspended in 500 µl PBS- and centrifuged again. This step was repeated. Each pellet was resuspended in 250 µl complete medium supplemented with DAPI at (0.05 µg/ml) and 0.5 µl of CellEvent Caspase 3/7 Green detection reagent (ex. 511/ em. 533) and incubated at 37 °C for 25 min. Cells were then sorted for mCherry (positive for transfection), DAPI and Caspase activity.

**Flow cytometry Annexin V.** One of the earlier events of apoptosis includes translocation of membrane phosphatidylserine (PS) from the inner side of the plasma membrane to the surface. Annexin V, a Ca$^{2+}$-dependent phospholipid-binding protein, has high affinity for PS, and fluorochrome-labeled Annexin V can be used for the detection of exposed PS. Cells were transfected as described above. Supernatant and cells were recuperated as described above. Cell pellets were resuspended in 100 µl of 1X binding buffer (10 mM Hepes/NaOH pH7.4, 140 mM NaCl, 2.5 mM CaCl$_2$). A concentration of 5 µl of Annexin V (Annexin-V-FLUOS, Roche) and DAPI (0.05 µg/ml) were added and the sample was incubated for 15 min at RT. A concentration of 200 µl of binding buffer was added before sorting. Cells were then sorted for mCherry (positive for transfection), DAPI and AnnexinV binding.

**LDH assay.** LDH activity was determined using Cytotoxicity detection kit (LDH) (Roche 11644793001). 50 000 cells were seeded in a 24-well plate with and without 1 µg/ml DOX. The next day, cells were treated either with vehicle (DMSO) or TM (1 µg/ml). After 16 h, 250 µl of medium from each well was removed and centrifuged at 250 × g for 10 min. A concentration of 100 µl of supernatant was transferred in a 96-well flat bottom microplate in duplicate and 100 µl of reaction mixture was added to each well. After 30 minutes of incubation at room temperature, absorbance was measured at 492 nm for LDH activity and at 600 nm for background control (subtracted to the average value of the duplicate at 492 nm). The percentage cytotoxicity was calculated according to the manufacturer's instructions.

**Lysosomal enzyme activity assays.** PCT cells were seeded into black-walled clear-bottom 96-well plates (Grienier Bio-One) at a density of 3 × 10$^4$ cells per well, in the presence or absence of 1 µg/ml DOX. Cells were cultured for 96 h as described above. Pharmacological manipulations were made 8 h prior to testing enzyme activities. For extracellular activity of lysosomal enzymes, substrates were added to wells containing intact cells and conditioned cell culture media. Before measuring cytosolic activity or total intracellular activity of lysosomal enzymes of interest, cells were washed in PBS, and the plasma membranes of the cells were permeabilized using 20 µg/ml digitonin, or cells were completely lysed using 200 µg/ml digitonin, respectively. Substrates were diluted into potassium phosphate buffer, pH 5.5, and added to wells containing either permeabilized or lysed cells. Cathepsin B and L activity was detected using the fluorogenic cathepsin substrate Z-Phe-Arg-AFC (Enzo Life Sciences), $\lambda_{ex}$ 310 ± 10 nm, $\lambda_{em}$ 500 ± 10 nm. N-acteyl-beta-D-glucosaminidase (NAG) activity was detected using the fluorogenic substrate, 4-Methylumbelliferyl N-acetyl-β-D-glucosaminidide (Sigma Aldrich), $\lambda_{ex}$ 365 ± 10 nm, $\lambda_{em}$ 445 ± 10 nm. Cleavage of fluorogenic substrates by lysosomal enzymes of interest was monitored using a Tecan Infinite M1000 Pro plate reader. Substrate cleavage was monitored for 90 minutes at 37°C, and peak RFU values were used for analysis of enzyme activity.

**TUNEL assay.** Terminal deoxynucleotidyl transferase (TdT)-mediated 2′-deoxyuridine 5′-triphosphate-biotin nick-end labeling (TUNEL) staining was performed by using the In Situ Cell Death Detection Kit (Roche, Mannheim, Germany). Labeling of 3′-OH terminal DNA fragments was then performed at 37 °C for 1 h by using the TUNEL reaction mixture according to the manufacturer's protocol.

Mouse renal sections were rehydrated and incubated with Proteinase K for 15 min at 37 °C. Tissue auto fluorescence was quenched using TrueBlack®

Lipofuscin Autofluorescence Quencher. Tissue was surrounded with a hydrophobic barrier using a barrier pen and incubated with TUNEL reaction mixture for 1 h at 37 °C in a humidified atmosphere in the dark. Sections were double stained with Hoechst 33342 (Molecular Probe, H3570, at 50 µg/ml for 10 min at room temperature) to visualize all the nuclei in the field. The sections were observed under a Zeiss Videomicroscope (Axiovert200M). Percentage of cell death was determined on stitched image of the whole kidney section.

A total of 1500 cells were seeded in a 48-well plate with and without 1 µg/ml DOX. The next day, cells were treated either with vehicle (DMSO) or TM (1 µg/ml). After 16 h, cells were fixed with 4% w/v PFA in PBS (1 h) and permeabilized (2 min on ice) with 0.1% TritonX-100 in 0.1% sodium citrate. Cells were then incubated with TUNEL reaction mixture for 1 h at 37 °C in a humidified atmosphere in the dark. Cells were washed twice in PBS and incubated with Hoescht for 10 min at RT. Samples were analyzed on a Zeiss microscope with a ×20 objective (Axioplan2 Imaging) using an excitation wavelength at 488 nm. Images were processed and quantified.

**Histology.** Kidneys were dissected, decapsulated and fixed in 4% paraformaldehyde in DPBS (Dulbecco's Phosphate Buffered Saline; Gibco BRL Life Technologies) for 24 h. Fixed tissues were dehydrated in graded ethanol and xylene and embedded in paraffin. Sequential sections of 5 µm were mounted onto Superfrost-plus glass slides. After deparaffinization, sections were stained with hematoxylin and eosin. Tubular lesions were quantified using the Image J software.

**Chemicals.** All chemicals were obtained from Sigma-Aldrich, except for Fura-2, AM from Molecular Probes (Invitrogen, France). Culture mediums were purchased from Invitrogen, France. FBS was purchased from Thermo Scientific HyClone.

**Metabolic parameters.** Procedures to measure metabolic parameters in mice were previously described[63].

**Electrophysiological measurements of ER-enriched liposomes.** ER microsomes from PCTs enriched with PC2 and eGFP or PC2 and TMEM33 were fused to lipid bilayers[7,10]. Experiments were performed with 250 mM HEPES-Tris solution, pH 7.35 on the *cis* and 250 mM HEPES, 55 mM Ba(OH)$_2$ solution, pH 7.35 on the *trans* side. Divalent cation concentrations on the cytoplasmic side of the PC2 channel were maintained by adjusting the ratio of calcium and EGTA. PC2 channels were activated by increasing calcium on the *cis* side. Channel activity was recorded for 2 min at each calcium concentration under voltage-clamp conditions on a Bilayer Clamp BC-525C (Warner Instuments, Hartford, CT, USA), filtered at 1 kHz and digitized at 5 kHz. Data was acquired and analyzed with pClamp9 (Axon Instruments, Burlingame, CA).

**Zebrafish experiments.** Zebrafish strains: zebrafish were maintained according to institutional and national ethical and animal welfare guidelines. All experiments were performed under UK Home Office licenses 40/3708 and 70/8588. The zebrafish lines *Tg(-26wt1b:EGFP)*[li1 64] and *tmem33*[sh443 43] were used.

Zebrafish morpholino injections: Embryos were injected at the 1 cell stage. A morpholino complementary to the ATG region of zebrafish *pkd2* was injected at 1 ng (5'-AGGACGAACGCGACTGGAGCTCATC-3')[42]. A concentration of 0.4 ng or 1 ng control morpholino (5'- CCTCTTACCTCAGTTACAATTTATA-3') was injected in these experiments.

Image acquisition and analysis: Zebrafish embryos were imaged at 52–55hpf when cysts became visible, using a Leica M165FC stereo microscope and images were acquired using Leica LASX software. Images were quantified using FIJI v1.52i. Zebrafish renal phenotype analysis was double blinded.

**Statistical analysis.** When appropriate, significance of the differences was tested with a permutation test (R Development Core Team: http://www.r-project.org/) ($n < 30$) or with two samples $t$ test ($n > 30$). One star indicates $p < 0.05$, two stars $p < 0.01$ and three stars $p < 0.001$. Data represent mean ± standard error of the mean. The number of independent times each experiment has been repeated ($n$) is indicated throughout the manuscript and is shown in figures or indicated in the text. The number of mice (N) used is also indicated throughout the figures.

**Ethical issues.** Experiments were carried out in accordance with the guidelines of the national institutional ethical committee for experimental animals and conform to the European community standards for the care and use of laboratory animals. All manipulations involving animals were carried out under controlled laboratory conditions by qualified personnel. The procedure followed for mouse euthanasia was in strict accordance with the European community standards on the care and use of laboratory animals. Animals were obtained from government-approved animal-raising companies, and a register was kept to record the origin of each animal, all its movements within the laboratory, and the reason for death. This study was approved by our local Committee for ethical and safety issues (CIEPAL-Azur).

**Reporting summary**. Further information on research design is available in the Nature Research Reporting Summary linked to this article.

## Data availability

Raw data for Figs. 1a, b; 2; 3a, b; 3d; 4a–g; 5a–f; 6; 7; 8; Supplementary Figs. 1a, b, d-f; 2,e, 2f, 2h, 2j; 3,a-d; 4,d-f; 5, b; 6a-f; 9,a, b, e, f and 10 are provided as a Source Data file. All data are available from the corresponding author upon reasonable request.

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

## Acknowledgements

We are grateful to Dr. Steve Somlo for providing us with the PCT *Pkd2*−/− cells and to Dr. Valérie Doye for sharing with us the Hela cell line stably expressing GFP-TMEM33. We deeply thank Dr. Fréderic Brau, Julie Cazareth and Sophie Abelanet for their technical assistance with cell imaging and sorting. We are grateful to the Fondation pour la Recherche Médicale (Equipe FRM to EH), as well as to the Agence Nationale de la Recherche (Project TMEM33 to EH), as well as to NIH (GM088790 to GSG, JSM) for funding this work.

## Author contributions

A.P. initiated this study. M.A., C.E.B. and F.D. performed Fura-2 fluoresence experiments on cultured PCT cells. G.G. performed cathepsins/NAG translocation experiments and quantification of electron microscopy micrographs. I.K. performed ER liposome bilayer experiments. C.M., M.P., H.D., N.P., S.D. and A.P. performed in vivo experiments. D.L., H.D., M.A., A.C. and S.D. performed qPCR measurements. C.M., M.P., M.A., A.P. and A.C. performed Western blots. A.P., M.A. and N.P. did the metabolic measurements. C.D. and I.R. generated immortalized WT and *TMEM33* KO PCT cells. S.P., S.L.G. generated electron microscopy micrographs. A.M.S., F.J.M.V.E., R.N.W. performed the zebrafish experiments. M.A., A.P., E.H. and G.G. generated the figures. E.H. wrote the manuscript. B.E.E., J.S.M. and A.P. edited the manuscript. This work was co-supervised by E.H. and A.P.

## Additional information

**Competing interests:** The authors declare no competing interests

