## [Peer Review File · Nature Communications]

Reviewers' comments:

Reviewer #1 (Remarks to the Author):

This study analyzes the interaction between TMEM33 and the ADPKD protein PC2. An interaction of the two proteins on the ER membrane is demonstrated along with the role that TMEM33 plays in regulating PC2 activity. Knockout studies of TMEM33 show a potential role in ER stress and possible significance for AKI. The study includes careful analysis of the role of TMEM33 in regulating the PC2 channel using a combination of knockdown and knockout reagents. The whole animal studies are also interesting, providing strong data on the protective role that loss of TMEM33 can have following induced AKI.

Specific points

1. The interaction between PC2 and TMEM33 seems complex with N terminal and C-terminal regions involved. This is not well explained.
2. Was the wider phenotype of the Tmem33 knockouts analyzed? From the data presented, the presence of TMEM33 just seems to be detrimental.
3. Histological images showing the lack of cystic phenotype in older Tmem33 knockouts should be illustrated.

Reviewer #2 (Remarks to the Author):

In this study, Arhatte et al. reveals the interaction of TMEM33 with polycystin-2. Interestingly, the interaction of TMEM33 appears to regulate PC2 in terms of calcium channel gating in ER. In addition, evidence is shown that TMEM33 is associated with endolysosomal calcium refilling and cathepsin translocation. In mice, TMEM33 knockout protected against tunicamycin-induced nephrotoxicity or acute kidney injury.

The study by Arhatte et al. has several original findings including TMEM33/PC2 interaction, regulation of PC2 Ca²⁺ channel gating, endolysosomal refilling etc. These findings add significantly to the understanding of TMEM and related cellular responses.

However, some of the conclusions need more solid evidence to support.

1. Most data are from overexpression and silencing. It is important to document the evidence of endogenous TMEM33.
2. In Fig. 1A, where is the IgG-IP control? Without appropriate controls, it is difficult to judge the Co-IP data.
3. For yeast two-hybrid study in Fig. 1C. Is this system specifically used for membrane protein study? Why did not the authors include the full-length TMEM33 and PC2? In the Sakebe paper, they proposed the topology of TMEM33 with N-term in ER lumen and C-term in cytoplasm, while PC2 is a membrane protein with N- and C-terminus all in the cytoplasm. How do the authors reconcile all these results with your findings in yeast two hybrid?
4. HeLa cell is not an optimal cell model for studying TMEM33 and PC2 co-localization, so it is important to verify the key results by using kidney cells. In addition, ER marker should be added in Fig. 1D, Fig. Suppl 1B and D. In Fig. 1E, TMEM33 and PC2 seem overexpressed? An antibody to TMEM33 should be used to determine the co-localization with endo-PC2, ER marker should be added the same time. Why are the staining patterns so similar in Fig. 1E? The general consensus is that very few molecules of PC2 can go to cilia if overexpressed alone. PC1 may facilitate PC2 translocation to cilia. Similarly, in co-IP study, does endo-TMEM33 interact with endo-PC2 at ER?
5. In Fig. 1F, what are the TMEM33 protein level in different tissues? Considering TM-induced kidney injury, it would be interesting to know how TMEM33 is expressed in different kidney cells in control and TM-treated mouse kidneys.
6. The authors studied the effect of TMEM33 on PC2. TRPV4 is also a calcium channel and interacts with PC2. Is TRPV4 affected?

7. The authors measured intracellular calcium with Fura-2 AM in some experiments. What is the Y-axis unit?

8. The authors did not provide TMEM33 protein level after Pkd2 knockout in Fig. Suppl 2A and B. Mild to moderate knockdown of one gene at transcript level cannot guarantee efficient knockdown at protein level. In Fig. 2E, authors studied the effect of thapsigargin on TMEM33-HA, how about endogenous TMEM33?

9. Calcium experiments were performed with transient overexpression systems. Transfection rate usually significantly affect the results. What are the transfection rates for PCT cells and Pkd2 cells? Are PCT cells characterized and published?

10. In Fig. 4, Pkd2 and/or TMEM33 knockdown cells were used for experiments. Western blot is missing for the knockdown. It is true for many other experiments, either overexpression or knockdown. If a protein is knocked down (or knockout) or overexpressed, it is necessary to verify by Western blots.

11. In Fig. Suppl 4A and B, are endo-TMEM33 and endo-PC1/2 colocalized?

12. Is endo-TMEM33 expressed in lysosome? An overexpressed protein can traffic aberrantly. Thus, to check the endo-TMEM33 in lysosome is necessary. Is endogenous PC2 in lysosome?

13. Is the TMEM33 mouse line characterized and published? If not, the authors need to add the information to the methods, including the knockout strategy and genotyping. Does the knockout generate no protein, partial truncation, or mutation etc.? It is better to provide the evidence of knockout.

Other issues.

1. In a few figure panels, texts are not well organized and labelled. For instance, in Fig. 1A, it is confusing what is IP and what is IB? In Fig. Suppl 1C, one blot was not labelled. Is PC2 IB missing? The authors need to pay attention to this.

2. It is better for all the antibodies used to be listed in one table or one paragraph. Information for each antibody should be provided, including source and catalog number, because one company may generate multiple antibodies against the same protein.

Reviewer #3 (Remarks to the Author):

This study by Arhatte and colleagues employed mainly cell models and also mouse models to study physical and functional interactions of TMEM33 and PC2. They found that TMEM33 enhances the channel function (by electrophysiology and Ca measurements) of PC2 on ER membrane, thereby decreasing the steady state level of calcium on the ER lumen, as well as ATP-elicited IP3-dependent Ca leak to the cytoplasm and the capacitive Ca entry from the extracellular space. Likely through this and their further studies they found that the TMEM33/PC2 complex also regulates apoptosis and endolysosomal structure/function. Using TMEM33 KO mice they claimed that they established link between the TMEM33/PC2 complex and AKI. In general, this study revealed important new functions of TMEM33 or the TMEM33/PC2 complex.

Major comments.

1. Although it is very interesting to see the protective effect of TMEM33 KO on mouse survival, apoptosis, tubular dilation and AKI following TM treatment, and although the data on apoptosis using TMEM33 KO mice are consistent with those using PCT cells, the studies using these mice remained rather descriptive, mainly because they do not have data on the effect of PC2 on AKI. Therefore, the last sentence in Abstract, "...establish a link between the TMEM33/PC2 complex and acute kidney injury" is an overstatement. Consequently, at this stage, the manuscript title has to be revised accordingly.

In fact, the most important disease implication of PC2 is that the abnormal PC2 function (or dosage), notably reduced PC2 function/dosage results in ADPKD. But contributions of abnormal function of PC2 at PM, cilia or ER membrane to ADPKD remain debatable. Thus, this study should examine whether TMEM33 has protective effect on PC2-dependent cystogenesis, ie specifically, whether TMEM33 KO (that decreases PC2 function) aggravates cystic diseases caused by PKD2+/- (or in PKD2-/- embryonic kidneys). If the authors believe that ER PC2 has no role in ADPKD (as TMEM is not found in primary cilia with PC2, Fig. 1E), then their data from this experiment would show that TMEM33 will not alter PC2-associated cyst progression/severity, which will represent very important data with respect to the mechanism of ADPKD.

2. The reviewer is not convinced by the model shown in Suppl Fig 9 and the associated concept of 'amplifier and leak modes'. Both IP3 and TMEM33 enhance PC2 channel's Ca transport, and the direction of Ca flow will be determined solely by the Ca electrochemical driving force across ER membrane. Thus, in both scenarios (with or without TMEM33), PC2 does not seem to have an amplifying (what is amplified?) effect.
3. What is known about the effect and mechanism of PC2 on cytotoxicity? The fact that TMEM33 enhances cytotoxicity suggests that PC2 itself should decrease cell viability (ie be pro cytotoxic); is it consistent with the literature? This would further suggest that dysfunction of PC2 due to pathogenetic mutations may be good for cell viability by reducing cytotoxicity. This appears surprising to me; any evidence of ADPKD patients benefit from their mutations in PC2, in terms of cytotoxicity? This should be discussed with existing publications, if any.
4. First paragraph of Results. "Of note, TMEM33 was also found at the nuclear membrane, unlike PC2 (Fig. 1D and Supp 1B)". The reviewer is not convinced by the shown data: Theoretically, when scanning section is through the nucleus, there should not be any staining for blue or red on the nucleus. Practically, staining within certain depth (layer) centered at the scanning section will contribute to the image. The scanning sections/layers across the nucleus for the two different wave lengths are not exactly the same and thus would distinctly bring in the 'contaminated' staining from above or underneath the nucleus.
5. Fig. 3C. Need statistical data (open probability and p value) to show the TMEM effect.
6. Fig. 4C and D. Need data to show the effectiveness of PC2 knockdown (as they did it for TMEM33); is there any effect of PC2 knockdown on TMEM33 expression?
7. Fig. 5C and D. Why using annexin (control)? Needs justification.
8. Fig. 6E and F. Needs statistics for TUNEL data.

Minor comments.

1. Second page in Discussion. "The ER voltage is clamped at 0 mV, because of a large basal K⁺-selective conductance governed by TRIC channels and an equal K⁺ concentration on both sides of the ER membrane". This to me appears to be contradictory, because the cytoplasm usually has negative voltages, eg -50 mV. Based on the statement of 'predominant K⁺ permeability and equal K⁺ concentration across the ER membrane', ER lumen should have the same negative voltage value, ie -50 mV, rather than 0 mV. The next sentence seems to be problematic too, "Whether PC2 acts a genuine ER calcium release channel by its own or whether it fulfills the role of a counterion channel (permeating K⁺) is unknown at this stage". Assume that ER PC2 is also a K⁺ channel, then it should see the same electrochemical driving force as the TRIC channel, ie, it won't be able to act as a counterion channel on the same membrane. Other places in the manuscript using the word 'counterion' might also need to be revised accordingly.
2. Fig. 1D. Use of an ER marker protein would be of help.

3. TM treatment dose and time are crucial with respect to resulting ER stress or apoptosis. Also, the two factors should be different between cell and animal models. Need justifications for the use of the two factors.
4. Page numbers should be included in the manuscript.
5. Results, section "TMEM33 sensitizes PCT cells to apoptosis". Not sure whether 'data not shown' is allowed. 'TM' was not defined when first time used.
6. Fig. 4C and D. Not sure whether the peaks differences are statistically different. The y-axis scales are different, which reduces the actual differences. Should show p value.
7. Needs full name for TMEM33.
8. Justification for using detergent CL76.
9. Line 10 in the first paragraph of Results. Needs to add start and end amino acid numbers for "N/C-terminus of TMEM33 and N/C terminus of PC2" Are they the whole N/C terminus or just partial?

Reviewer #1 (Remarks to the Author):

This study analyzes the interaction between TMEM33 and the ADPKD protein PC2. An interaction of the two proteins on the ER membrane is demonstrated along with the role that TMEM33 plays in regulating PC2 activity. Knockout studies of TMEM33 show a potential role in ER stress and possible significance for AKI. The study includes careful analysis of the role of TMEM33 in regulating the PC2 channel using a combination of knockdown and knockout reagents. The whole animal studies are also interesting, providing strong data on the protective role that loss of TMEM33 can have following induced AKI.

Thank you for your enthusiastic comments.

Specific points

1. The interaction between PC2 and TMEM33 seems complex with N terminal and C-terminal regions involved. This is not well explained.

We have now better explained the predicted topology of TMEM33 and detailed this complex molecular interaction. Page 4, lines 7-9.

2. Was the wider phenotype of the Tmem33 knockouts analyzed? From the data presented, the presence of TMEM33 just seems to be detrimental.

We have now explored the role of TMEM33 in renal cystogenesis (novel Fig. 8). Page 9.

3. Histological images showing the lack of cystic phenotype in older Tmem33 knockouts should be illustrated.

We have now added illustration to demonstrate the lack of cysts older TMEM33^{-/-} mice (Fig. Supp 8).

Reviewer #2 (Remarks to the Author):

In this study, Arhatte et al. reveals the interaction of TMEM33 with polycystin-2. Interestingly, the interaction of TMEM33 appears to regulate PC2 in terms of calcium channel gating in ER. In addition, evidence is shown that TMEM33 is associated with endolysosomal calcium refilling and cathepsin translocation. In mice, TMEM33 knockout protected against tunicamycin-induced nephrotoxicity or acute kidney injury.

The study by Arhatte et al. has several original findings including TMEM33/PC2 interaction, regulation of PC2 Ca²⁺ channel gating, endolysosomal refilling etc. These findings add significantly to the understanding of TMEM and related cellular responses.

We are grateful to this reviewer for his positive comments and constructive suggestions.

However, some of the conclusions need more solid evidence to support.

We have now provided multiple new control and validation experiments to better support our study. We have added new Fig. 1D, Fig. 8, Fig. 9, Fig. Supp 1A, Supp 1D, Fig. Supp 2C-J, Fig. Supp 5, Fig. Supp 8, Fig. Supp 9C and D.

1. Most data are from overexpression and silencing. It is important to document the

evidence of endogenous TMEM33.

We agree that this is an important point. We screened several commercially available antibodies directed against TMEM33, as well as one antibody produced by HKU antibody facility (see below). None of these antibodies gave us a specific signal, either comparing WT and TMEM33^{-/-} PCT cells or TMEM33^{-/-} PCT cells complemented cells with HA tagged TMEM33 constructs. One antibody against TMEM33 was previously published by Sakabe and collaborators¹. We requested this reagent, but unfortunately we were told that this antibody is not anymore available for a test. So, at this stage we could not identify a validated anti-TMEM33 that would allow us to faithfully analyze native TMEM33. To overcome this difficulty, we decided to take advantage of our stably complemented TMEM33^{-/-} PCT cells expressing either HA-TMEM33 or TMEM33-HA that shows levels of expression (as determined by qPCR), comparable to native TMEM33 for validation of siRNAs at the protein level, in addition to our previous qPCR findings. Fig. Supp 2E-J. Moreover, these tagged TMEM33 complemented TMEM33^{-/-} cells allowed us to study co-localization with native PC2 at the endoplasmic reticulum (we identified the Santa Cruz E20 antibody that labels specifically PC2; Fig. Supp1G).

2. In Fig. 1A, where is the IgG-IP control? Without appropriate controls, it is difficult to judge the Co-IP data.

We have now provided a control experiment using IgG. No immunoprecipitation between TMEM33 and PC2 is observed with IgG, unlike with the specific anti-HA antibody (Fig. Supp 1A). In addition, in Fig 1 A, while there is no IgG control, anti-HA immunoprecipitation of Myc-PC2 or MYC-PC2742X does not occur in the absence of HA-TMEM33, thus confirming a specific interaction. The same holds for Fig. 1B wherein Myc-TMEM33 is not anti-HA immunoprecipitated in the absence of HA-PC2.

3. For yeast two-hybrid study in Fig. 1C. Is this system specifically used for membrane protein study? Why did not the authors include the full-length TMEM33 and PC2? In the Sakebe paper, they proposed the topology of TMEM33 with N-term in ER lumen and C-term in cytoplasm, while PC2 is a membrane protein with N- and C-terminus all in the cytoplasm. How do the authors reconcile all these results with your findings in yeast two hybrid?

The yeast two-hybrid system used relies on a molecular interaction within the nucleus (i.e. valid for cytosolic fragments, but not for whole transmembrane proteins). This is now clearly indicated in the method section. Various topology models predict that both N and C termini of human TMEM33 and its yeast homologues are facing the cytosol^{2, 3, 4}. Those reports are now cited in the revised manuscript (Page 4, lines 7-9).

4. HeLa cell is not an optimal cell model for studying TMEM33 and PC2 co-localization, so it is important to verify the key results by using kidney cells.

We have replaced the HeLa data by experiments performed in renal PCT cells (Fig. 1D, Fig. Supp 5A). In addition, ER marker should be added in Fig. 1D, Fig. Suppl 1B and D. We have now used ER tracker and calnexin as markers of ER (Fig. 1D and Fig. Supp 5A). In Fig. 1E, TMEM33 and PC2 seem overexpressed? We have now clearly indicated that we overexpress TMEM33-GFP and mCherry-PC2 in transfected PCT cells. An antibody to TMEM33 should be used to determine the co-localization with endo-PC2, ER marker should be added the same time. Since we lack a specific AB against TMEM33 (see above), we have used complemented TMEM33^{-/-} cell expressing HA-TMEM33 at physiological level (Fig. Supp 5A). In this experiment we detect co-localization with native PC2 (E20 antibody from Santa Cruz) at the ER, using calnexin as an ER marker. Why are the staining patterns so similar in Fig. 1E? They are similar as the same cells are visualized. TMEM33 is detected by green GFP fluorescence, acetylated-tubulin as a marker of the primary cilium in far red and PC2 in red. False colors were then used to generate the images in Fig. 1E. The general consensus is that very few molecules of PC2 can go to cilia if overexpressed alone. PC1 may facilitate PC2 translocation to cilia. Recent findings from the Clapham group cast doubt on this targeting mechanism⁵. PC2 is localized to the primary cilium in Pkd1^{-/-} cells. A targeting signal to the primary cilium was localized within the N terminal region of PC2⁶. Similarly, in co-IP study, does endo-TMEM33 interact with endo-PC2 at ER? We now show in stably complemented TMEM33^{-/-} PCT cells that both HA-TMEM33 and TMEM33-HA interact with native PC2 (Fig. Supp 1A).

5. In Fig. 1F, what are the TMEM33 protein level in different tissues? Unfortunately, we were told that the previously published TMEM33 antibody is not available anymore¹.

Considering TM-induced kidney injury, it would be interesting to know how TMEM33 is expressed in different kidney cells in control and TM-treated mouse kidneys.

We took advantage of the LacZ reporter to compare TMEM33 expression in the cortex and

medulla of mice treated either with the vehicle or TM (now shown in Fig. Supp 9C-D).

6. The authors studied the effect of TMEM33 on PC2. TRPV4 is also a calcium channel and interacts with PC2. Is TRPV4 affected? We now show that TRPC4 expression is not different between WT and KO mice either treated with the vehicle or with TM (Fig. Supp 2D).

7. The authors measured intracellular calcium with Fura-2 AM in some experiments. What is the Y-axis unit? This is a ratio of fluorescence $\Delta F/R_0$, thus there is no unit, as now explained in the method.

8. The authors did not provide TMEM33 protein level after Pkd2 knockout in Fig. Suppl 2A and B. Mild to moderate knockdown of one gene at transcript level cannot guarantee efficient knockdown at protein level. We now show in Fig. Supp 2F-J the efficiency of TMEM33 and PC2 knock-down by siRNAs at the protein level. In Fig. 2E, authors studied the effect of thapsigargin on TMEM33-HA, how about endogenous TMEM33? It was previously reported that endogenous TMEM33 in mammary cancer cells is strongly induced by ER stress¹. Unfortunately, the antibody used is not anymore available (see above). In our complemented cell line, we failed to observe an increase in TMEM33 expression at the protein level, even with higher doses of TM or thapsigargin. These negative findings suggest that some regulatory elements are probably lacking in the PCT clone.

9. Calcium experiments were performed with transient overexpression systems. Transfection rate usually significantly affect the results. What are the transfection rates for PCT cells and Pkd2 cells? Transfection efficiency is now indicated in the method. Page lines. Are PCT cells characterized and published? PCT cell lines were previously published by⁷ and this reference is cited in the method. Page 6, top.

10. In Fig. 4, Pkd2 and/or TMEM33 knockdown cells were used for experiments. Western blot is missing for the knockdown. It is true for many other experiments, either overexpression or knockdown. If a protein is knocked down (or knockout) or overexpressed, it is necessary to verify by Western blots. We have now provided validation of the siRNAs both at the mRNA and protein levels (Fig. Supp 2F-J).

11. In Fig. Suppl 4A and B, are endo-TMEM33 and endo-PC1/2 colocalized? Taking advantage of TMEM33^{-/-} stably complemented cells (level of expression comparable to the native TMEM33 as determined by qPCR), we now show that TMEM33 is co-localized with endogenous PC2 (E20 AB) at the ER (visualized with calnexin AB)(Fig. Supp 5A).

12. Is endo-TMEM33 expressed in lysosome? An overexpressed protein can traffic aberrantly. Thus, to check the endo-TMEM33 in lysosome is necessary. Is endogenous PC2 in lysosome? We found evidence for localization of native PC2 and HA-TMEM33 within lysosomes (Fig. Supp 5B-C). However, since ER-phagy occurs, it is possible that both proteins might derive from ER membranes engulfed in autolysosomes.

13. Is the TMEM33 mouse line characterized and published? If not, the authors need to add the information to the methods, including the knockout strategy and genotyping. Does the knockout generate no protein, partial truncation, or mutation etc.? It is better to

provide the evidence of knockout. We have now included this information that was missing in the revised method section. Page 4, bottom.

Other issues.

1. In a few figure panels, texts are not well organized and labelled. For instance, in Fig. 1A, it is confusing what is IP and what is IB? In Fig. Suppl 1C, one blot was not labelled. Is PC2 IB missing? The authors need to pay attention to this.

Sorry about mislabeling the initial figures. This has now been improved in the revised version (Fig. 1A-B and Fig. Supp 1A). It is better for all the antibodies used to be listed in one table or one paragraph. Information for each antibody should be provided, including source and catalog number, because one company may generate multiple antibodies against the same protein. We have now provided a table in the revised method section with all requested information about antibodies used in this study. Page 8.

Reviewer #3 (Remarks to the Author):

This study by Arhatte and colleagues employed mainly cell models and also mouse models to study physical and functional interactions of TMEM33 and PC2. They found that TMEM33 enhances the channel function (by electrophysiology and Ca measurements) of PC2 on ER membrane, thereby decreasing the steady state level of calcium on the ER lumen, as well as ATP-elicited IP3-dependent Ca leak to the cytoplasm and the capacitive Ca entry from the extracellular space. Likely through this and their further studies they found that the TMEM33/PC2 complex also regulates apoptosis and endolysosomal structure/function. Using TMEM33 KO mice they claimed that they established link between the TMEM33/PC2 complex and AKI. In general, this study revealed important new functions of TMEM33 or the TMEM33/PC2 complex.

We are grateful to this reviewer for his positive comments and very helpful input.

Major comments.

1. Although it is very interesting to see the protective effect of TMEM33 KO on mouse survival, apoptosis, tubular dilation and AKI following TM treatment, and although the data on apoptosis using TMEM33 KO mice are consistent with those using PCT cells, the studies using these mice remained rather descriptive, mainly because they do not have data on the effect of PC2 on AKI. Therefore, the last sentence in Abstract, "...establish a link between the TMEM33/PC2 complex and acute kidney injury" is an overstatement. We accordingly deleted "PC2" from this sentence. Consequently, at this stage, the manuscript title has to be revised accordingly. We have modified the title and now indicate activation of ER PC2 by TMEM33. Effects of TMEM33 on AKI is indicated in the abstract. In fact, the most important disease implication of PC2 is that the abnormal PC2 function (or dosage), notably reduced PC2 function/dosage results in ADPKD. But contributions of abnormal function of PC2 at PM, cilia or ER membrane to ADPKD remain debatable. Thus, this study should examine whether TMEM33 has protective effect on PC2-dependent cystogenesis, ie specifically, whether TMEM33 KO (that decreases PC2 function) aggravates cystic diseases caused by PKD2+/- (or in PKD2-/- embryonic kidneys). If the authors believe that ER PC2 has no role in ADPKD (as TMEM is not found in primary cilia with PC2, Fig. 1E),

then their data from this experiment would show that TMEM33 will not alter PC2-associated cyst progression/severity, which will represent very important data with respect to the mechanism of ADPKD.

We agree it is a key point that is now addressed in the revised manuscript. We reasoned that the potential role of TMEM33 needs to be investigated in a PC2 hypomorphic model (that recapitulates the disease mechanism), as we would not anticipate to see a protective effect of TMEM33 (activator of ER PC2) if the channel is absent (i.e. *Pkd2*^{-/-}). Implementing a mouse *Pkd2* hypomorphic model in the *TMEM33*^{-/-} background represents a major technical issue and could not be achieved within the time frame allowed for this revision. To overcome this major difficulty, we took advantage of a validated knock-down strategy of *pkd2* in zebrafish. This model nicely recapitulates the molecular mechanism (dosage) and main clinical feature of ADPKD culminating in renal cysts (Fig. 8). Using this approach, we found no evidence for a protective role of *tmem33* against *pkd2*-dependent renal cystogenesis (Fig. 8). We conclude from this data that ADPKD is most probably independent of ER PC2.

2. The reviewer is not convinced by the model shown in Suppl Fig 9 and the associated concept of ‘amplifier and leak modes’. Both IP3 and TMEM33 enhance PC2 channel’s Ca transport, and the direction of Ca flow will be determined solely by the Ca electrochemical driving force across ER membrane. Thus, in both scenarios (with or without TMEM33), PC2 does not seem to have an amplifying (what is amplified?) effect. We have simplified this part of the discussion and removed the “amplification” statement in the text, as well as in the model.

3. What is known about the effect and mechanism of PC2 on cytotoxicity? The fact that TMEM33 enhances cytotoxicity suggests that PC2 itself should decrease cell viability (ie be pro cytotoxic); is it consistent with the literature? This would further suggest that dysfunction of PC2 due to pathogenetic mutations may be good for cell viability by reducing cytotoxicity. This appears surprising to me; any evidence of ADPKD patients benefit from their mutations in PC2, in terms of cytotoxicity? This should be discussed with existing publications, if any.

The situation is rather complex as PC2 at different subcellular localizations (primary cilium, ER, cell junction, nucleus...) affects multiple signaling pathways that might differentially affect cell survival, proliferation, toxicity and apoptosis. Interestingly, recent studies indicate that PC2 stimulates autophagy in a variety of cell types, including renal epithelial cells, as well as cardiomyocytes, involving both the primary cilium and intracellular calcium stores^{8, 9, 10, 11}. Autophagy, that is activated by ER stress, is generally considered to be cytoprotective^{12, 13}. Since our findings suggested a role for ER TMEM33/PC2 in the regulation of lysosomal function, we further investigated a possible impact of TMEM33 on autophagy. We observed that TMEM33 stimulates the autophagic flux in basal conditions (with an expected cytoprotective action), while it inhibits autolysosomes degradation (with a consequent accumulation of LC3II) upon TM-induced ER stress (loss of protection). We have now illustrated this dual opposite effect on autophagy in a revised model (Fig. 9). We propose that the combination of TMEM33-dependent lysosomal cathepsins translocation and impaired autophagic flux promotes cell death upon ER stress.

4. First paragraph of Results. “Of note, TMEM33 was also found at the nuclear membrane, unlike PC2 (Fig. 1D and Supp 1B)”. The reviewer is not convinced by the shown data: Theoretically, when scanning section is through the nucleus, there should not be any staining for blue or red on the nucleus. Practically, staining within certain depth (layer)

centered at the scanning section will contribute to the image. The scanning sections/layers across the nucleus for the two different wave lengths are not exactly the same and thus would distinctly bring in the 'contaminated' staining from above or underneath the nucleus. **We have accordingly deleted this statement.**

5. Fig. 3C. Need statistical data (open probability and p value) to show the TMEM effect. **Significance of the differences is indicated with stars on top of the histogram.**

6. Fig. 4C and D. Need data to show the effectiveness of PC2 knockdown (as they did it for TMEM33); is there any effect of PC2 knockdown on TMEM33 expression? **We have now provided several validation experiments demonstrating at the protein level the effectiveness of both PC2 and TMEM33 knock down by transfection of siRNAs (Fig. Supp 2F-J). Knock down of TMEM33 does not affect PC2 expression (Fig. Supp 2F). Moreover, in the TMEM33^{-/-} cell line stably complemented with HA-tagged TMEM33, knock down of Pkd2 does not affect TMEM33 protein expression (Fig. Supp 2J).**

7. Fig. 5C and D. Why using annexin (control?)? Needs justification. **Calnexin is an integral protein of the ER that runs at about 90 kDa and was used as loading control as its molecular weight allows a convenient separation on the same gel with TMEM33 (about 28 kDa). Since both PC2 and TMEM33 are ER proteins, it is appropriate to use another transmembrane ER protein as a loading control (<https://www.labome.com/method/Loading-Controls-for-Western-Blots.html>). We now indicate in the legend of Fig. Supp 2 the use of calnexin as a loading control for this Western blot.**

8. Fig. 6E and F. Needs statistics for TUNEL data. **In Fig. 7F, significance of the differences is shown by stars on top of the histogram.**

Minor comments.

1. Second page in Discussion. "The ER voltage is clamped at 0 mV, because of a large basal K⁺-selective conductance governed by TRIC channels and an equal K⁺ concentration on both sides of the ER membrane". This to me appears to be contradictory, because the cytoplasm usually has negative voltages, eg -50 mV. Based on the statement of 'predominant K⁺ permeability and equal K⁺ concentration across the ER membrane', ER lumen should have the same negative voltage value, ie -50 mV, rather than 0 mV. The next sentence seems to be problematic too, "Whether PC2 acts a genuine ER calcium release channel by its own or whether it fulfills the role of a counterion channel (permeating K⁺) is unknown at this stage". Assume that ER PC2 is also a K⁺ channel, then it should see the same electrochemical driving force as the TRIC channel, ie, it won't be able to act as a counterion channel on the same membrane. Other places in the manuscript using the word 'counterion' might also need to be revised accordingly.

We have deleted this part of the discussion for simplicity.

2. Fig. 1D. Use of an ER marker protein would be of help.

ER markers (ER tracker, as well as calnexin) are now shown in Fig. 1D and Fig. Supp 5A.

3. TM treatment dose and time are crucial with respect to resulting ER stress or apoptosis. Also, the two factors should be different between cell and animal models. Need justifications for the use of the two factors. **For *in vitro* experiments, we used previously**

published effective dose (1 µg/ml)^{14, 15}. For *in vivo* experiments, in preliminary test experiments we compared the effect of 1 and 2 mg/Kg^{15, 16}. At 1 mg/Kg no consistent effect on ER stress marker (GRP78 and CHOP) expression was observed. Thus, we chose the dose of 2 mg/kg as it gave us highly reproducible ER stress effect and the dose was sublethal in the TMEM33 KO line.

4. Page numbers should be included in the manuscript. **Done.**

5. Results, section “TMEM33 sensitizes PCT cells to apoptosis”. Not sure whether ‘data not shown’ is allowed. ‘TM’ was not defined when first time used. **Corrected.**

6. Fig. 4C and D. Not sure whether the peaks differences are statistically different. The y-axis scales are different, which reduces the actual differences. Should show p value. **Significance is now shown by stars.**

7. Needs full name for TMEM33. **This is now indicated in the abstract.**

8. Justification for using detergent CL76. **In our initial study we used both CL76 and CL90 for mass spectrometry analysis. Since the initial publication is cited¹⁷, we deleted this technical information.**

9. Line 10 in the first paragraph of Results. Needs to add start and end amino acid numbers for “N/C-terminus of TMEM33 and N/C terminus of PC2” Are they the whole N/C terminus or just partial? **We have now inserted this information in the text, as well in the methods.**

References cited:

1. Sakabe I, Hu R, Jin L, Clarke R, Kasid UN. TMEM33: a new stress-inducible endoplasmic reticulum transmembrane protein and modulator of the unfolded protein response signaling. *Breast Cancer Res Treat* 153, 285-297 (2015).
2. Chadrin A, *et al.* Pom33, a novel transmembrane nucleoporin required for proper nuclear pore complex distribution. *J Cell Biol* 189, 795-811 (2010).
3. Zhang D, Oliferenko S. Tts1, the fission yeast homologue of the TMEM33 family, functions in NE remodeling during mitosis. *Mol Biol Cell* 25, 2970-2983 (2014).
4. Zhang D, Vjestica A, Oliferenko S. The cortical ER network limits the permissive zone for actomyosin ring assembly. *Curr Biol* 20, 1029-1034 (2010).
5. Liu X, Vien T, Duan J, Sheu SH, DeCaen PG, Clapham DE. Polycystin-2 is an essential ion channel subunit in the primary cilium of the renal collecting duct epithelium. *Elife* 7, (2018).
6. Geng L, *et al.* Polycystin-2 traffics to cilia independently of polycystin-1 by using an N-terminal RVxP motif. *J Cell Sci* 119, 1383-1395 (2006).
7. L'Hoste S, *et al.* Role of TASK2 in the control of apoptotic volume decrease in proximal kidney cells. *J Biol Chem* 282, 36692-36703 (2007).
8. Orhon I, *et al.* Primary-cilium-dependent autophagy controls epithelial cell volume in response to fluid flow. *Nat Cell Biol* 18, 657-667 (2016).
9. Criollo A, *et al.* Polycystin-2-dependent control of cardiomyocyte autophagy. *J Mol Cell Cardiol* 118, 110-121 (2018).

10. Pena-Oyarzun D, *et al.* Hyperosmotic stress stimulates autophagy via polycystin-2. *Oncotarget* 8, 55984-55997 (2017).
11. Lu J, *et al.* Polycystin-2 Plays an Essential Role in Glucose Starvation-Induced Autophagy in Human Embryonic Stem Cell-Derived Cardiomyocytes. *Stem Cells* 36, 501-513 (2018).
12. Yang Y, Ma F, Liu Z, Su Q, Liu Y, Li Y. The ER-localized Ca(2+)-binding protein calreticulin couples ER stress to autophagy by associating with microtubule-associated protein 1A/1B light chain 3. *J Biol Chem* 294, 772-782 (2019).
13. Levine B, Kroemer G. Biological Functions of Autophagy Genes: A Disease Perspective. *Cell* 176, 11-42 (2019).
14. Dong G, Liu Y, Zhang L, Huang S, Ding HF, Dong Z. mTOR contributes to ER stress and associated apoptosis in renal tubular cells. *Am J Physiol Renal Physiol* 308, F267-274 (2015).
15. Zinszner H, *et al.* CHOP is implicated in programmed cell death in response to impaired function of the endoplasmic reticulum. *Genes Dev* 12, 982-995 (1998).
16. Huang L, *et al.* Increased susceptibility to acute kidney injury due to endoplasmic reticulum stress in mice lacking tumor necrosis factor-alpha and its receptor 1. *Kidney Int* 79, 613-623 (2011).
17. Sharif Naeini R, *et al.* Polycystin-1 and -2 dosage regulates pressure sensing. *Cell* 139, 587-596 (2009).

REVIEWERS' COMMENTS:

Reviewer #1 (Remarks to the Author):

I am happy that my concerns have now been addressed.

Reviewer #2 (Remarks to the Author):

In this revision, the authors have included new data to address previous concerns. Although there is no information about the endogenous protein due to the lack of a good antibody, the authors tried their best and provided the data of reconstitution cells.

Reviewer #3 (Remarks to the Author):

The revised manuscript has reflected the authors' responses to and incorporations of the referee's comments. In particular, the no effect of TMEM33 on PC2-dependent disease was shown using zebrafish (Fig. 8). The quality of the manuscript has been significantly improved. However, with all the changes made, something discordant appears between the title and the content:

The fact that TMEM33 does not regulate PC2 knockdown-induced zebrafish pronephric cyst formation indicates that the ER TMEM33/PC2 complex and ER PC2 are not directly related to cystogenesis. The question then is what is the physiological role of the TMEM33/PC2 complex and PC2 located on the ER membrane, which should be examined in an animal model. Because data on role of the TMEM33/PC2 complex or PC2 on autophagy, apoptosis, cytotoxicity, lysosomal structure/function etc were all from using in vitro cultured cells, they did not provide answer to the question. The only experiments using mice were about the role of TMEM33 on AKI, which is more or less irrelevant to the theme/title of this study which is about regulation of PC2 by TMEM33 and can be removed from the manuscript without sacrificing anything significant. But then the only in vivo data are those from zebrafish. In summary, the study, as stated by the title, would have to be documented by in vivo data.

Reviewer #1 (Remarks to the Author):

I am happy that my concerns have now been addressed.

We are grateful to this reviewer for his valuable input.

Reviewer #2 (Remarks to the Author):

In this revision, the authors have included new data to address previous concerns. Although there is no information about the endogenous protein due to the lack of a good antibody, the authors tried their best and provided the data of reconstitution cells.

We are glad that this reviewer was satisfied with our corrections and additions.

Reviewer #3 (Remarks to the Author):

The revised manuscript has reflected the authors' responses to and incorporations of the referee's comments. In particular, the no effect of TMEM33 on PC2-dependent disease was shown using zebrafish (Fig. 8). The quality of the manuscript has been significantly improved. However, with all the changes made, something discordant appears between the title and the content: The fact that TMEM33 does not regulate PC2 knockdown-induced zebrafish pronephric cyst formation indicates that the ER TMEM33/PC2 complex and ER PC2 are not directly related to cystogenesis. The question then is what is the physiological role of the TMEM33/PC2 complex and PC2 located on the ER membrane, which should be examined in an animal model. Because data on role of the TMEM33/PC2 complex or PC2 on autophagy, apoptosis, cytotoxicity, lysosomal structure/function etc were all from using in vitro cultured cells, they did not provide answer to the question. The only experiments using mice were about the role of TMEM33 on AKI, which is more or less irrelevant to the theme/title of this study which is about regulation of PC2 by TMEM33 and can be removed from the manuscript without sacrificing anything significant. But then the only in vivo data are those from zebrafish. In summary, the study, as stated by the title, would have to be documented by in vivo data.

We have accordingly toned down our title and related conclusions concerning the role of PC2.